# ADDRESSING THE STABILITY-PLASTICITY DILEMMA VIA KNOWLEDGE-AWARE CONTINUAL LEARNING

## ABSTRACT

Continual learning agents should incrementally learn a sequence of tasks while satisfying two main desiderata: accumulating on previous knowledge without forgetting and transferring previous relevant knowledge to help in future learning. Existing research largely focuses on alleviating the *catastrophic forgetting* problem. There, an agent is altered to prevent forgetting based solely on previous tasks. This hinders the balance between preventing forgetting and maximizing the forward transfer. In response to this, we investigate the *stability-plasticity* dilemma to determine which model components are eligible to be reused, added, fixed, or updated to achieve this balance. We address the class incremental learning scenario where the agent is prone to ambiguities between old and new classes. With our proposed Knowledge-Aware contiNual learner (KAN) [1], we demonstrate that considering the semantic similarity between old and new classes helps in achieving this balance. We show that being aware of existing knowledge helps in: (1) increasing the forward transfer from similar knowledge, (2) reducing the required capacity by leveraging existing knowledge, (3) protecting dissimilar knowledge, and (4) increasing robustness to the class order in the sequence. We evaluated sequences of similar tasks, dissimilar tasks, and a mix of both constructed from the two commonly used benchmarks for class-incremental learning; CIFAR-10 and CIFAR-100.

## 1 INTRODUCTION

Continual learning (CL) aims to build intelligent agents based on deep neural networks that can learn a sequence of tasks, use previous knowledge in future learning, and accumulate on it without forgetting. The main challenge in this paradigm is the stability-plasticity dilemma (Mermillod et al., 2013). Optimizing all model weights on a new task, *highest plasticity*, causes catastrophic forgetting of previous tasks (McCloskey & Cohen, 1989). While fixing all weights, *highest stability*, hinders learning new tasks. Finding the right balance between stability and plasticity is challenging. This sharpens the community's focus on the forgetting problem. The excessive focus on one aspect impedes building agents that balance between mitigating forgetting and exploiting relevant knowledge in future learning while considering the capacity constraints (Díaz-Rodríguez et al., 2018).

Task-specific components methods (Section 2) address the stability-plasticity dilemma by allocating different connections for each task. The newly added connections for a new task are flexible to learn while the connections of previously learned tasks are fixed. Despite that these methods are quite successful in mitigating forgetting, some limitations remain to be tackled to achieve the balance between the above-mentioned CL desiderata. *First*, new components (connections/neurons) are allocated in *each layer* for every new task. However, the new components might capture knowledge that already exists in the learned components of previous *similar* tasks. In this case, adding new components would be redundant and resource inefficient. *Second*, the topology of the new task is allocated using the *unimportant* components of previous tasks. The design choice of the new topology is made to protect previous knowledge but does not take into consideration the usefulness of this topology for the current task. This limits the forward transfer of useful knowledge for the new task and its speed of learning. Further discussions of these limitations are in Appendix A.

---

[1] The code will be made publicly available after the publication of this paper and now can be found in the supplementary material in the system.

In response to these limitations, we study the core of the stability-plasticity dilemma in the CL paradigm. In particular, we address the following question: *Which components are eligible to be reused, added, updated, or fixed when a CL agent faces a new task to achieve the balance between the CL desiderata?* With our proposed Knowledge-Aware contiuNal learner (KAN), we find that considering the semantic similarity between old and new classes is crucial in addressing this question. With the awareness of existing knowledge, the CL agent could identify *similar* previous knowledge that could be reusable in learning a new task (forward transfer) and add the necessary components only to capture the specific knowledge that cannot be explained with the existing knowledge (resource efficiency). Moreover, the agent could protect the *dissimilar* previous knowledge and limit its transfer to the new task (mitigating forgetting).

To the best of our knowledge, the recent work, CAT (Ke et al., 2020), is the only one that attempted to consider the task similarities to balance between forward transfer and mitigating forgetting. CAT was proposed for the *task-Incremental Learning* (task-IL) scenario, where each task is a separate classification problem. In this work, we aim to tackle the more challenging *class-Incremental Learning* (class-IL) scenario where a unified classifier is used for all seen classes. The inaccessibility of the task identity at inference in the latter scenario arises the following challenges: (1) ambiguities between old and new tasks which affect the balance between maintaining forgetting and forward transfer. (2) Inability to select the corresponding components to a task at inference; all the existing components in the model are involved in making predictions. To fully analyze the stability-plasticity dilemma in class-IL, given the aforementioned challenges, we focus on the rehearsal-free strategy (Section 2) in which there is no reliance on past data to address forgetting and a *fixed-capacity* model is used. We also report the performance of the model in Task-IL to further demonstrate the challenges in class-IL. Our contributions are:

- We demonstrate that considering the semantic relation between old and new classes is crucial in addressing the stability-plasticity dilemma and the balance between CL desiderata.

- We show that considering both previous and current tasks in allocating the *initial* topology of a new task allows for: selective transfer for the relevant (similar) knowledge, protecting the irrelevant (dissimilar) knowledge, and identifying the reusable previous components to avoid allocating unnecessary resources.

- We demonstrate that the standard softmax-classification layer is a source of constituting the ambiguities between old and new tasks in class-IL. We study different setups for the output layer which significantly improve the performance of rehearsal-free methods. Our analyses reveal the limitations of the softmax in class-IL and shed light on the necessity of paying more attention to these limitations to narrow the gap between class-IL and task-IL.

- We propose Knowledge-Aware coNtinual leaner (KAN); a resource-efficient task-specific components method to explain these findings.

## 2 RELATED WORK

We divide the CL methods into two main categories: Rehearsal-free and Rehearsal-based methods.

**Rehearsal-free methods.** In this category, previous data is inaccessible during future learning. There are two strategies to address forgetting. **(1) *Task-specific components.*** *Specific* components are assigned to each task. The components of previous tasks are *stable* during future learning. While the components that have not been allocated to any task are *flexible* for future learning. These methods either *extend* the model size when facing a new task (Rusu et al., 2016; Yoon et al., 2018) or uses a *fixed-capacity* and each task is trained either using *sparse* sub-network within the model (Mallya & Lazebnik, 2018; Mallya et al., 2018; Yoon et al., 2019). These methods are limited to task-IL as they require the task identity to specify the components of each task at inference. Recently, few methods have been proposed for class-IL using *fixed-capacity* model. SupSup (Wortsman et al., 2020) learns a mask for each task over *randomly initialized fixed* network. FSLL (Mazumder et al., 2021) addressed the few-shot lifelong learning setup. It relies on the availability of large data for the first task to train a dense model. Then, few unimportant parameters are used to learn each task. SpaceNet (Sokar et al., 2021c) learns sparse sub-network for each task from scratch using dynamic sparse training (Mocanu et al., 2018; Hoefler et al., 2021) where the weights and the sparse topology are optimized simultaneously. **(2) *Regularization-based.*** A *fixed-capacity* model is used and

*all* parameters are involved in optimizing each task. Forgetting is addressed either by constraining changes in the important weights of previous tasks (Zenke et al., 2017; Kirkpatrick et al., 2017; Aljundi et al., 2018) or via distillation loss (Li & Hoiem, 2017; Dhar et al., 2019). Previous studies (Kemker et al., 2018; Hsu et al., 2018; Farquhar & Gal, 2019; van de Ven & Tolias, 2018) showed the big performance gap of most of these methods between class-IL and task-IL.

**Rehearsal-based methods.** In this category, forgetting is addressed by *replaying*: (1) a subset of old samples (Rebuffi et al., 2017; Lopez-Paz & Ranzato, 2017; Riemer et al., 2018; Chaudhry et al., 2018), (2) pseudo-samples from a generative model (Mocanu et al., 2016; Shin et al., 2017; Sokar et al., 2021a), or (3) generative high-level features (Liu et al., 2020; van de Ven et al., 2020). They use a classification model and buffer to store old samples or a generative model to generate them.

**Other related studies.** Recent work by (Ramasesh et al., 2020) evaluates forgetting in hidden representations in Task-IL and shows how semantic similarities influence it. They showed that higher layers are more prone to forgetting and intermediate similarity results in maximal forgetting. SAM (Sokar et al., 2021b) studies the importance of selective transfer in dense networks by meta-training (Finn et al., 2017) a self-attention mechanism (Hu et al., 2018). CAT (Ke et al., 2020) addresses the relation between task similarities and forward/backward transfer in task-IL where they rely on the task identity to find the previous similar knowledge in dense networks. In (Chen et al., 2020), the lottery ticket hypothesis (Frankle & Carbin, 2018) is studied for the CL paradigm.

# 3 PRELIMINARIES

## 3.1 PROBLEM FORMULATION

We consider the problem of learning a sequence of $T$ tasks. Each task brings a new set of $C$ classes. Each task $t$ has its own dataset $\mathcal{D}^t = \{D_c^t\}|_{c=1}^C$, where $D_c^t$ is the data of class $c$ in task $t$. Once we have trained task $t$, its data is discarded. The goal is to sequentially learn the model $f^t$ with a unified classifier that can map any input to its corresponding target for all seen classes up to time step $t$.

## 3.2 REHEARSAL-FREE STRATEGY

In this work, we focus to analyze how the model components should be altered for each task in task-specific components methods. In addition, we compare the behavior of these methods against regularization-based methods which use all model components for each task.

### 3.2.1 SPACENET

We analyze the recent task-specific components method, SpaceNet (Sokar et al., 2021c). The motivations for choosing SpaceNet are: (1) unlike most task-specific component methods, it does not rely on task identity; making it applicable for class-IL. (2) Instead of relying on a pre-trained model, it trains sparse subnetworks from scratch. These are favorable for real-world applications in which we cannot assume the availability of task identity at inference and large datasets for pretraining. To the best of our knowledge, it is only existing work that fulfills these objectives.

The main idea of SpaceNet is to learn *sparse representation* for each task to reduce the interference between tasks. SpaceNet consists of three main steps. (1) Sparse connections allocation: sparse sub-network $W_l^t$ is *randomly* allocated for each task $t$ in *each* layer $l$ between subset of the *unimportant* neurons $\mathbf{h}_{l-1}^{sel}$ and $\mathbf{h}_l^{sel}$; where $\mathbf{h}_l^{sel}$ are randomly selected from the *free* neurons $\mathbf{h}_l^{free}$ that have not been reserved for any previous task. (2) Dynamic sparse training: the sparse weights and topology are optimized through "drop-and-grow" cycles to produce sparse representation. (3) Neuron reservation: after learning, a fraction of the most important neurons for the current task $\mathbf{h}_l^{spec}$ are fixed and removed from the *free* list of neurons $\mathbf{h}_l^{free}$. The full details are provided in Appendix I. SpaceNet considers only previous tasks during the allocation of new connections for the new tasks and randomly allocates a new sub-network between the free neurons for each task. *In this work, we propose novel criteria which consider both past and current tasks in allocation and allow reusing some of the past components instead of allocating new ones.*

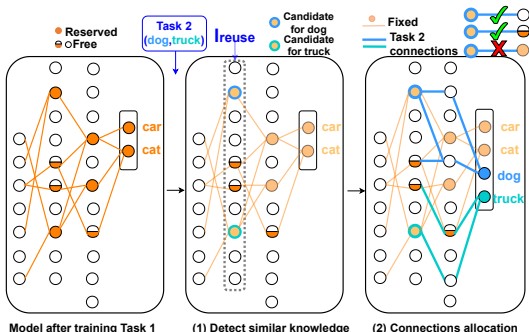

Figure 1: An overview of our proposed method KAN. The most left panel shows the network after training on Task 1. When the model faces Task 2, KAN reuses the existing similar knowledge up to layer $l_{reuse}$ and adds new components in the subsequent layers. For $l \geq l_{reuse}$, (1) the candidate similar neurons for each new class are detected. (2) New sparse connections are allocated between the candidate neurons and free neurons in layer $l-1$ and the free neurons in layer $l$.

## 4 PROPOSED KNOWLEDGE-AWARE CONTINUAL LEARNER (KAN)

In this work, we aim to study how the model components should be altered, when a CL agent faces a new task, to balance between the CL desiderata (i.e. forward transfer, reducing forgetting, and resource efficiency). To address this goal, we propose Knowledge-Aware contiNual learner (KAN), a new task-specific components model that exploits the semantic similarity between old and new classes in altering a CL model. The novel contributions of KAN are: (1) the allocation of a new topology considers both previous and current tasks; (2) reusing some of the existing learned components from previous similar tasks instead of allocating new components in *each layer*.

Figure 1 shows an overview of KAN. When the agent faces a new task $t$, new connections are allocated for this task. The new connections are trained with stochastic gradient descent (SGD). All previous connections are fixed. Instead of allocating new connections in each layer, KAN starts allocation from layer $l_{reuse} \in [1, L-1]$ which is a hyper-parameter that controls the trade-off between allocating new components and reusing old components based on existing knowledge. $L$ is the number of layers. The topology from layer $l_1$ up to but excluding layer $l_{reuse}$ remains unchanged. In KAN, we operate on the class level instead of tasks for connections allocation. Starting from $l_{reuse}$ onwards, KAN allocates new connections for each new class. This requires two steps.

**The first step** is the detection of similar knowledge. Typically, some of the previously learned classes are similar to the new classes and some are dissimilar. Classes that have semantic similarity are most likely to share similar representation (Appendix A; Figures 5b and 6). For each new class $c$, KAN finds the *candidate* neurons $\mathbf{h}_l^{c\text{-}cand}$ in each layer $l$, $\forall l \geq l_{reuse}$, that could contain *relevant* representation learned by a previous similar class. To achieve this, we feed the data of class $c$, $\mathcal{D}_c^t$, on the trained model at time step $t-1$, $f^{t-1}$, and calculate the average activation $\mathbf{a}_l^c$ in each layer $l$. We estimate the relevance by the activation value. The higher the activation is, the more relevant is the neuron (i.e. the neurons that fire are the most important to the current class). We select a fraction $sel_l^{c\text{-}cand}$ of the most relevant neurons to be the candidate neurons $\mathbf{h}_l^{c\text{-}cand}$ for class $c$. Please note that by exploiting the representation learned in the candidate neurons $\mathbf{h}_{l_{reuse}}^{c\text{-}cand}$ in layer $l_{reuse}$, we are *reusing* previous learned components connected to these neurons in preceding layers ($l < l_{l_{reuse}}$). Therefore, these components are stable but reusable.

**The second step** is to exploit the relevant candidate neurons in each layer $l$, $\forall l \geq l_{reuse}$, in allocating the new connections for each new class $c$. Adding new connections in layer $l$ between the candidate neurons $\mathbf{h}_{l-1}^{c\text{-}cand}$ and $\mathbf{h}_l^{c\text{-}cand}$ could maximize the forward transfer from previous knowledge. However, this would lead to forgetting previously similar knowledge as the representation learned by the important neurons for previous classes would be updated. To solve this dilemma, we allow reusing the candidate neurons by allocating new connections *out* of these neurons. However, we do not allow new connections to be added *into* the *important* neurons for previous tasks $\mathbf{h}_l^{spec}$ (fully filled circles in Figure 1). In addition, we block the gradient flow through all important neu-

---

**Algorithm 1** KAN Connections Allocation

---

1: **Require:** $sel_l^t$, $sel_{l-1}^{c\text{-}cand}$, $l_{reuse}$, sparsity level $\epsilon_l$
2: **for each** class $c$ in task $t$ **do**                    ▷ Get candidate neurons in each layer for each new class
3:     $\mathbf{a}_l^c \leftarrow$ forward pass $\mathcal{D}_c^t$ on $f^{t-1}$
4:     $\widetilde{\mathbf{a}}_l^c \leftarrow$ sortDescending $(\mathbf{a}_l^c)$
5:     $\mathbf{h}_l^{c\text{-}cand} \leftarrow$ select the first $sel_l^{c\text{-}cand}$ from $\widetilde{\mathbf{a}}_l^c$
6: **end for each**
7: $(\mathbf{h}_{l-1}^{sel}, \mathbf{h}_l^{sel}) \leftarrow$ randomly select $sel_{l-l}$ and $sel_l^t$ neurons from $\mathbf{h}_{l-1}^{free}$ and $\mathbf{h}_l^{free}$
8: **for each** class $c$ in task $t$ **do**
9:     **for** $l \leftarrow l_{reuse}$ to $L-1$ **do**
10:         $\mathbf{h}_{l-1}^{alloc} \leftarrow \{\mathbf{h}_{l-1}^{sel} \cup \mathbf{h}_{l-1}^{c\text{-}cand}\}$
11:         $\mathbf{h}_l^{alloc} \leftarrow \mathbf{h}_l^{sel}$
12:         randomly allocate parameters $W_l^c$ with sparsity $\epsilon_l$ between $\mathbf{h}_{l-1}^{alloc}$ and $\mathbf{h}_l^{alloc}$
13:         $W_l \leftarrow W_l \cup W_l^c$
14:     **end for each**
15: **end for each**

---

rons for previous tasks $\mathbf{h}_l^{spec}$, even if it is a candidate neuron for the current class, to protect previous knowledge. Thus, $\mathbf{h}_l^{spec}$ is used to mask the gradient $g_l$ through the neurons in layer $l$ as follows: $g_l = g_l \otimes (1 - \mathbf{h}_l^{spec})$, where $\mathbf{h}_l^{spec}$ is a one-hot encoding vector in which the entities contain value of one represent an important neuron in layer $l$. *This allows us* not to forget the previously learned knowledge while performing selective transfer for the useful knowledge. Let $\mathbf{h}_{l-1}^{alloc}$ and $\mathbf{h}_l^{alloc}$ be the list of neurons for allocating new sparse connections $W_l^c$ for class $c$ in layer $l$ with a sparsity level $\frac{\epsilon_l}{C}$; where $\epsilon_l$ is a hyper-parameter that controls the number of sparse connections for each task and $C$ is the total number of classes. The neurons used for allocation are selected as follows:

$$\mathbf{h}_{l-1}^{alloc} = \{\mathbf{h}_{l-1}^{cand} \cup \mathbf{h}_{l-1}^{sel}\} \quad \text{and} \quad \mathbf{h}_l^{alloc} = \mathbf{h}_l^{sel}, \quad \text{where} \quad \mathbf{h}_i^{sel} \in \mathbf{h}_i^{free} \quad \forall i = \{l, l-1\}, \quad (1)$$

where $\mathbf{h}_i^{sel}$ is a subset, $sel_i^t$, of the free neurons in layer $i$ that are not reserved for any previous task. Since the important neurons for the dissimilar classes are not involved in the topology allocation, we are protecting their knowledge. Thus, the selective transfer in KAN addresses the forward transfer while maintaining previous similar and dissimilar knowledge. Full details are in Algorithm 1.

After allocating the topology for task $t$, we follow steps 2 and 3 of SpaceNet, dynamic sparse training and neuron reservation, to train the task and fix its important neurons $\mathbf{h}_l^{spec}$ afterwards (Section 3.2.1).

## 5 EXPERIMENTS AND RESULTS

We now use KAN to investigate how the model should deal with sequences of similar, dissimilar tasks, or a mix of both to achieve the balance between the competing CL desiderata. We compare against other rehearsal-free methods. We analyze the accuracy, forward and backward transfer, model capacity, and class-order sensitivity. Each experiment is repeated 5 times with different seeds.

### 5.1 DATASETS AND BASELINES

**Datasets.** We construct sequences with varying similarities from the standard benchmarks for class-IL, CIFAR-10 and CIFAR-100 (Zenke et al., 2017), by altering the original class order. *From CIFAR-10*, we construct two-tasks and five-tasks sequences. For the two-tasks sequences, we construct: (1) Similar sequence (**sim_seq_2Tasks**): Task1{car,cat}, Task2{dog,truck}, where the two tasks share semantic similarity. (2) Dissimilar sequence (**dissim_seq_2Tasks**): Task1{car,truck}, Task2{cat,dog}, where the semantic similarity is limited. For the five-tasks sequence (**sim_seq_5Tasks:**), we arranged the 10 classes of CIFAR-10 in the following order:{car,cat,horse,truck,dog,deer,airplane,bird,ship,truck}. Each task consists of two consecutive classes. With this order, we study the case where there is past knowledge that has common semantic similarity with the new task. *From CIFAR-100*, we construct a longer sequence of 8 tasks (**sim_seq_8Tasks**). Each task contain two classes coming from two different categories and each two consecutive tasks shares semantic similarity. The classes has the following order: {apples,girl, orange,boy,mouse,bicycle,rabbit,motocycle,bee,lion,butterfly,tiger,bottle,couch,chair,cans}.

Table 1: Forward transfer. Test accuracy [%] of Task 2 in two cases: similar and dissimilar tasks.

| Strategy | Method | sim_seq_2Tasks | dissim_seq_2Tasks | # Parameters allocated |
|---|---|---|---|---|
| Use whole capacity | SI $c_{reg} = 0.1$ | $98.19 \pm 0.240$ | $76.05 \pm 3.95$ | $1\times(884576)$ |
| Use whole capacity | SI $c_{reg} = 0.01$ | $98.30 \pm 0.127$ | $81.91 \pm 1.60$ | $1\times(884576)$ |
| Use one sub-network | Scratch | $97.73 \pm 0.008$ | $80.21 \pm 0.02$ | $(0.065 \times)$ |
| Add new sub-network per task | SpaceNet | $\mathbf{98.17} \pm 0.002$ | $\mathbf{81.27} \pm 0.02$ | $(\mathbf{0.131} \times)$ |
| Reuse relevant knowledge | KAN (ours) $l_{reuse} = L - 1$ | $96.29 \pm 0.006$ | $66.17 \pm 0.02$ | $(0.065 \times)$ |
| Reuse random knowledge | Random reuse $l_{reuse} = L - 1$ | $95.43 \pm 0.002$ | $64.60 \pm 0.01$ | $(0.065 \times)$ |

**Architecture.** We followed the architecture used by (Zenke et al., 2017; Sokar et al., 2021c) for a direct comparison. It also allows us to study the challenging case where the available capacity is limited. It consists of 4 convolutional layers (32-32-64-64 feature maps) and 2 feed-forward layers (512-$N$ neurons), where $N$ is the number of classes. More details are in Appendix B.

**Baselines.** We compare with the following baselines: (1) *Scratch*: learning a single task from scratch using a sparse neural network with the same sparsity level used for one task in the task-specific components baselines, (2) *SpaceNet* (Sokar et al., 2021a), (3) *SI* (Zenke et al., 2017), (4) *Random reuse*: a modified version of KAN in which the neurons used for allocating new connections are subset of the free neurons (i.e. no candidate neurons), and (5) *Irrelevant reuse*: a modified version of KAN in which the candidate neurons are the ones with the lowest activation. The last two baselines allow us to study the role of the initial sparse topology in the performance. *In our experiments*, we report the performance of our modified improved version of SI where a multi-headed output layer is used instead of single-headed (Appendix H.2). A comparison with other baselines (PackNet (Mallya & Lazebnik, 2018), EWC (Serra et al., 2018), MAS (Aljundi et al., 2018), and LWF (Li & Hoiem, 2017)) and analysis of forgetting can be found in Appendix G.

## 5.2 SELECTIVE KNOWLEDGE TRANSFER FROM PREVIOUSLY LEARNED KNOWLEDGE

### 5.2.1 ATTENTION TO RELEVANT KNOWLEDGE

In this paragraph, we address the question: *Is all previous knowledge always useful for the current task?* To answer this question, we begin by investigating the extreme case where new connections for a new task are added in the output layer only (i.e. $l_{reuse} = L - 1$). In this case, the minimum new capacity is allocated, and the maximum reusability and highest stability of existing knowledge occur. We performed this analysis on two-tasks sequences (sim_seq_2Tasks, dissim_seq_2Tasks). To estimate the forward transfer, we focus on the performance of Task 2 assuming the availability of the task identity at inference. Table 1 shows the accuracy of Task 2 using different methods.

**In the similar case**, SI and SpaceNet achieve a better performance than learning from scratch which indicates that the learned knowledge by Task 1 is useful for Task 2. Interestingly, despite that Task 2 is trained by learning the output layer only in KAN, it achieves a close performance to SI and SpaceNet. This performance is achieved using only 6.5 % and 50% of the capacity used by SI and SpaceNet respectively. KAN outperforms the Random reuse baseline which reveals the importance of selective transfer. **In the dissimilar case**, using SI with $c_{reg}$ of 0.1 limits the performance of Task 2 while reducing it to 0.01 gives the connections the flexibility to learn at the expense of forgetting Task 1 as we will discuss next. This shows that solving the stability-plasticity dilemma by regularization strategies is challenging when tasks are dissimilar. This is because the changes in the model weights after learning a dissimilar task are larger than learning a similar one (Appendix A). In KAN, allocating new connections in the last layer only is not sufficient to learn Task 2 since the existing knowledge is irrelevant. This experiment reveals that adding new components in earlier layers is essential when existing knowledge is irrelevant. However, if there is a semantic similarity between classes, with a *selective transfer*, we could learn new classes with minimum components.

To further analyze the importance of the selective transfer, we performed another experiment on sim_seq_2Tasks in which we fool KAN by making the $\mathbf{h}_{L-1}^{c\text{-}cand}$ of class dog in Task 2 equals to the candidate neurons of the dissimilar class in Task 1 (car). We named this baseline as *Inattentive learner*. Figure 2a shows the test accuracy of Task 2 along training. As illustrated, attention to irrelevant knowledge leads to lower initial performance, slower learning speed, and lower final accuracy.

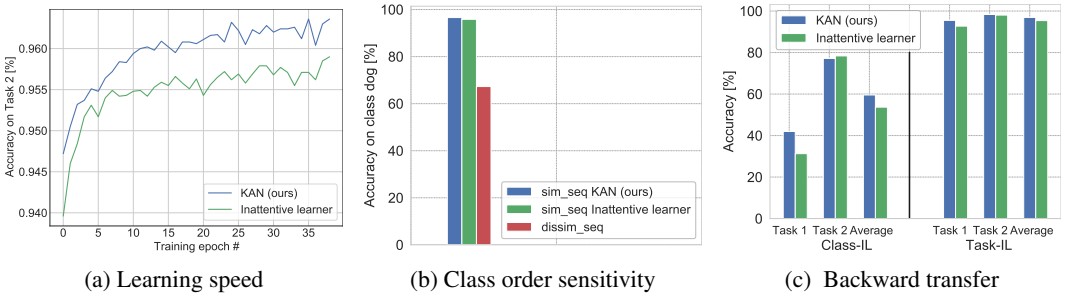

|              |              |             |
|:------------:|:------------:|:-----------:|
| (a) Learning speed | (b) Class order sensitivity | (c) Backward transfer |

Figure 2: (a) Using irrelevant knowledge decreases the learning speed. (b) Unawareness of existing knowledge decreases robustness to the class order. (c) Selective transfer reduces forgetting.

Previous experiments reveal the sensitivity of the learner to the class order when the relation between old and new classes is not considered or the irrelevant knowledge is used. Figure 2b shows the test accuracy of class dog which appears in Task 2 in three different settings: (1) Similar tasks with KAN, (2) Similar tasks with Inattentive learner (relevant knowledge exists but we reuse the dissimilar one), and (3) Dissimilar tasks with KAN (existing knowledge has limited relevance). Despite that the same class appears at the same time step in the three scenarios, its performance varies.

### 5.2.2 ADDING NEW COMPONENTS FOR A NEW TASK

We showed that new tasks could be learned using previous similar knowledge in Task-IL. *Does this hold in class-IL?* Next, we will discuss the *ambiguities* that arise in class-IL between old and new classes when they are similar. We performed this experiment on the sim_seq_2Tasks benchmark.

**Class ambiguities.** In class-IL, a unified classifier is used for all seen classes. Reusing all relevant knowledge from previous similar classes up to the highest representation level (closest layer to the output; $l_{reuse} = L - 1$) in learning a new class maximizes the forward transfer with minimum additional capacity. Nevertheless, it misleads the classifier and leads to ambiguities between old and new classes. The first row in Table 2, illustrates this. We report the accuracy of each task after learning the two. We find that in task-IL, we can easily fully utilize previous knowledge up to the last layer in learning a new task; achieving high performance on old and new tasks. However, in class-IL, the model tends to bias toward one of the tasks due to the ambiguity between classes. To address this problem, we propose two modifications for KAN: (1) usage of the free neurons only in allocating the output connections (i.e. no candidate neurons are used; $\mathbf{h}_{L-1}^{alloc} = \mathbf{h}_{L-1}^{sel}$). (2) Enforcing the re-usage of previous components to be till earlier layers than the highest one ($l_{reuse} < L - 1$) and adding new components in higher layers to capture the specific representations of the new classes. Next, we will study the effect of these proposed constraints. Ablation study is in Appendix D.

**Adding new components in earlier layer.** To reduce the ambiguities, we need to satisfy the condition $l_{reuse} < L - 1$. Here, we study the effect of $l_{reuse}$ in the performance. As illustrated in Table 2, adding new components in the last three layers, $l_{reuse} = L - 3$, achieves the highest balance between the two tasks for the two scenarios. Adding new components for Task 2 in earlier layers ($l_{reuse} < L - 3$) does not lead to a significant performance gain for Task 2 in Task-IL. Nevertheless, it consumes additional resources and hinders the balance between forgetting and forward transfer in class-IL. The learner becomes biased towards the second task and starts forgetting the previous one.

Figure 3 shows the performance on sim_seq_2Tasks using different methods. We compare the best two settings for KAN ($l_{reuse} = \{L - 2, L - 3\}$) against other baselines. SpaceNet achieves higher performance on Task 2 at the expense of forgetting Task 1. KAN achieves a balance between the two tasks and a bit higher average accuracy than SpaceNet in class-IL with less computational cost (Appendix E). Random re-usage of previous knowledge or paying attention to the least relevant one (irrelevant reuse) hinders learning the second task. The regularization-based method, SI, struggles in balancing between the two tasks in the class-IL. Using all the network parameters in learning each task and relying on the regularization coefficient $c_{reg}$ to address the stability and plasticity trade-off is challenging. The learner either maintains the performance of Task 1 using a high value of $c_{reg}$ or forgets it and is biased towards the second task by decreasing the value of $c_{reg}$.

Table 2: The accuracy [%] of each task in sim_seq_2Tasks after learning the two tasks in sequence along with their average accuracy using different values of $l_{reuse}$ in class-IL and task-IL.

| | Class-IL | | | Task-IL | | | |
|---|---|---|---|---|---|---|---|
| $l_{reuse}$ | **Task 1** | **Task 2** | **Average** | **Task 1** | **Task 2** | **Average** | **# Parameters allocated** |
| $L-1$ | 69.66 | 34.77 | 52.21 | 95.34 | 96.29 | 95.81 | $1\times(58145)$ |
| $L-2$ | 64.34 | 61.11 | 62.72 | 95.28 | 97.16 | 96.21 | $(1.85\times)$ |
| $L-3$ | **64.90** | **63.97** | **64.43** | **94.41** | **97.90** | **96.15** | $(1.93\times)$ |
| $L-4$ | 63.36 | 66.85 | 65.10 | 94.43 | 97.68 | 96.05 | $(1.97\times)$ |
| $L-5$ | 58.99 | 69.82 | 64.41 | 93.37 | 98.19 | 95.78 | $(1.99\times)$ |
| $L-6$ | 59.69 | 70.81 | 65.25 | 95.38 | 98.29 | 96.84 | $(2\times)$ |

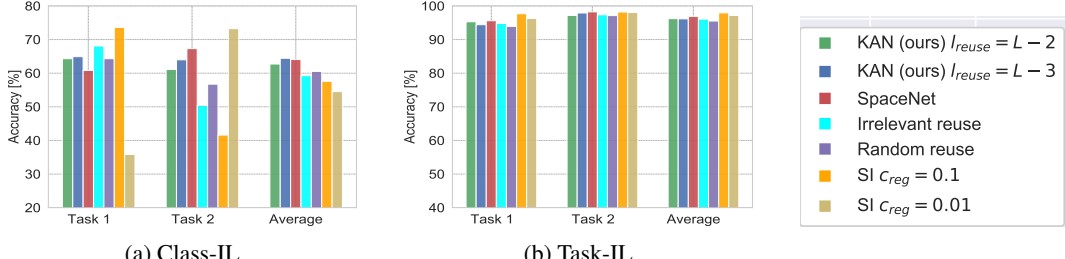

(a) Class-IL        (b) Task-IL

Figure 3: Test accuracy of each task in sim_seq_2Tasks and their average in class-IL and task-IL.

## 5.3 SELECTIVE BACKWARD TRANSFER FROM NEW TASKS

Constraining the capacity of the learner using a fixed-capacity model is one of the desiderata of CL. However, sooner or later this capacity would reach its limit and we have to use the existing connections to learn new tasks. In this experiment, we demonstrate that performing selective backward transfer based on the relevant knowledge reduces forgetting even if the previous weights are updated without regularization. We performed this experiment on sim_seq_2Tasks. We study the extreme case where new connections are added in the last layer only (i.e. $l_{reuse} = L-1$). We update the connections in all preceding layers which connected to the candidate neurons $\mathbf{h}_l^{c\text{-}cand}$ for $l: 2 \leq l \leq l_{reuse}$. We compare KAN with the Inattentive learner described in Section 5.2.1. Figure 2c shows the accuracy of each task in the sequence after learning the two and their average. With the selective transfer in KAN, we managed to maintain higher performance for Task 1 in class-IL and Task-IL. More interestingly, KAN outperforms the Inattentive learner by 10.4 % in class-IL. This reveals the importance of selective transfer for both *forward transfer* and *selective backward update*.

## 5.4 PERFORMANCE ON LONG SEQUENCE OF TASKS

Next, we study the performance of KAN on long sequences. For CIFAR-10 sim_seq_5Tasks, we use $l_{reuse} = L-3$. For CIFAR-100 sim_seq_8Tasks, since every two consecutive tasks are dissimilar to previous ones, we allocate new sparse connections in all layers to capture the new knowledge for every odd time step ($t = 1, 3, ..$). In other time steps, we use $l_{reuse} = L-3$. For SI, we use $c_{reg}$ of 0.01. As illustrated in Figure 4, KAN achieves higher performance than SpaceNet on the two benchmarks. The topology of the new connections matters; paying attention to relevant knowledge is better than using random neurons or irrelevant ones. SI is more prone to forgetting in long sequences of tasks. Further discussion on scalability and resource efficiency is in Appendix C.

## 5.5 SOFTMAX CLASSIFIER FOR CLASS-INCREMENTAL LEARNING

In this paper, we have observed the performance gap between class-IL and task-IL which was also reported by previous works (Maltoni & Lomonaco, 2019; Hsu et al., 2018). In our experiments, the learned representations by the base network in both scenarios are the same, this suggests investigating the output layer in class-IL. Current CL methods use the standard softmax. A single-head output layer is usually used and extended when new classes arrive. We named this as *Extendable layer*. The softmax has some limitations for this setup. It is based on the closed-world assumption (a

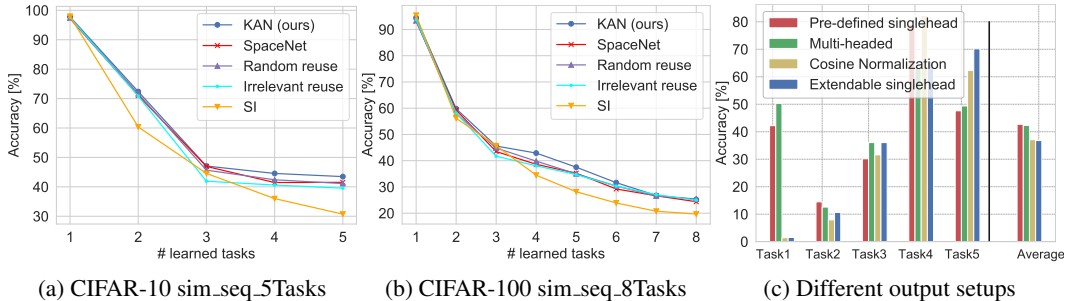

(a) CIFAR-10 sim_seq_5Tasks  (b) CIFAR-100 sim_seq_8Tasks  (c) Different output setups

Figure 4: The performance of different methods in class-IL on long sequences of tasks (CIFAR-10 (a) and CIFAR-100 (b)). (c) The performance of Split CIFAR-10 using different output layers.

fixed number of classes). It divides the feature space between known classes (Geng et al., 2020a;b); normalizing the output probabilities to be summed to one, leaving no space for future classes. In addition, the model is biased towards new classes due to the unavailability of old data (Wu et al., 2019). Few setups were proposed in the literature to alleviate these limitations: (1) *Cosine normalization* (Hou et al., 2019) which enforces balanced magnitudes of the output weights across all classes, (2) *Multi-headed layer* (Mittal et al., 2021) where each task has its own output layer, (3) *Pre-defined layer* (Sokar et al., 2021c) where a large number of output neurons is defined at $t = 0$. At each time step, the output weights of the current task are trainable and all other output weights are set to zero. The first two were evaluated for rehearsal-based methods. More details in Appendix H.

In this work, we evaluate these setups for the rehearsal-free strategy. We analyze the task-specific components method SpaceNet (Sokar et al., 2021c) on Split CIFAR-10 benchmark with the typical class order. As illustrated in Figure 4c, the commonly used extendable layer is the most prone to forgetting previous tasks. There is no significant gain by using cosine normalization with the extendable layer. Both the multi-headed and predefined single-headed, which address the closed-world assumption, succeed in mitigating forgetting while the latter achieves the best average performance. Analysis on the other rehearsal-free strategy, regularization-based, is provided in Appendix H.2.

## 5.6 CLASS SIMILARITY DETECTION: LEARNING CLASS RELATIONSHIP

In the previous analyses, we assume that we know whether the model previously learned a similar class to the new one. Automatic detection of class relationships is required. We evaluated the approach by (You et al., 2020) which averages the predictions of the model $f^{t-1}$ over all samples of each new class; $p(y_c^{t-1}|y_c^t) \approx \frac{1}{|\mathcal{D}_c^t|} \sum_{(x, y_c^t)} f^{t-1}(x)$. We observe that it works well in short sequences but does not scale for long sequences. Another approach by (Ke et al., 2020) is to compare the performance of task $t$ when it learns from scratch against learning from the model $f^{t-1}$. However, it is computationally expensive. In this work, we focus on analyzing the effect of considering task similarities. We leave designing sophisticated methods for similarity detection for future work.

## 6 CONCLUSION

In this paper, we study the stability-plasticity dilemma in continual learning where the model learns incrementally a sequence of tasks. We focus on the rehearsal-free methods using a fixed-capacity model in class-IL. With our proposed Knowledge-Aware contiNual learner (KAN), we analyzed how should the model be altered under different tasks similarity. Our analysis reveals that taking into consideration the relation between old and new classes is crucial to balance the CL desiderata. We show that it helps in: (1) identifying the existing similar knowledge which could be exploited to increase the forward transfer and reduce the allocated capacity for a new task; (2) protecting dissimilar knowledge via selective update. We demonstrate that achieving a balance between CL desiderata is more challenging in class-IL than in task-IL. Moreover, we show that the conventional softmax is a cause of this challenge. Our findings suggest that more efforts should be dedicated to (1) detecting the class similarities, (2) designing attention mechanisms for selective transfer, and (3) addressing the limitations of the softmax classifier to close the gap between class-IL and task-IL.

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

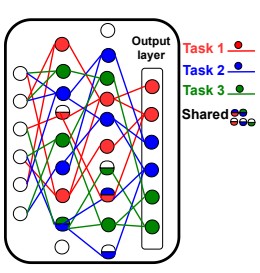 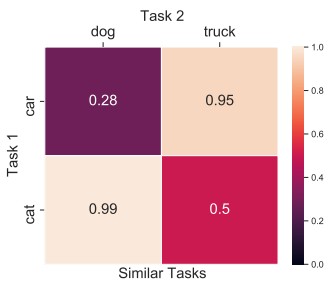 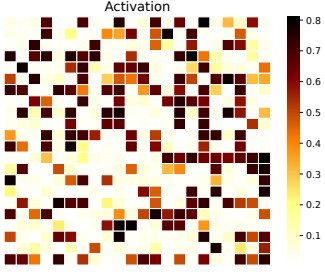

(a) New components/task      (b) Representational similarity      (c) Sparse representation

Figure 5: (a) New components are allocated per task in each layer. (b) Cosine similarity between the last hidden representation of each class in Task 1 and Task 2 produced by the model trained on Task 1. Task 2 has representational similarity with Task 1 that could be exploited in its learning. (c) Activation of the last hidden layer of one class in Task 2 using the *dense* model trained on Task 1. The relevant representation from previous knowledge is sparse.

## A  OBSERVATIONS IN THE CONTINUAL LEARNING PARADIGM

In this appendix, we will discuss some of the observations in the continual learning paradigm and provide a further discussion on the limitations stated in Section 1.

**New components of each task.** Figure 5a, show an illustration of connections allocation in the task-specific components method SpaceNet (Sokar et al., 2021c). New connections are added in each layer for each new task. The new connections are randomly allocated between the free neurons that could be shared between tasks. The full-filled colored neurons represent the neurons that are important for a task and reserved (fixed). The criteria for adding new connections consider only protecting previous knowledge.

**Representational similarities between tasks.** Figure 5b shows the representational similarities between classes in two similar tasks. This experiment is performed on the sim_seq_2Tasks benchmark using the same experimental setup for KAN described in Section 5.1 and Appendix B. The representations are obtained from the last hidden layer (closest to the output). We average the activation, $\mathbf{a}$, of the model trained on Task 1 $f^1$ over all samples of each class in Task 1 and Task 2. The cosine similarity between the two representations $\mathbf{a}_1, \mathbf{a}_2$ is calculated as follows: $< \bar{\mathbf{a}_1}, \bar{\mathbf{a}_2} > = \bar{\mathbf{a}_1}^T \bar{\mathbf{a}_2}$; where $\bar{\mathbf{a}_i} = \frac{\mathbf{a}_i}{||\mathbf{a}_i||}$ denotes the L2-normalized vector. As illustrated in the figure, when the classes share some semantic similarity, as the case in this benchmark, there would be representational similarity between the classes. In KAN, we exploit this in learning Task 2 by maximizing the forward transfer from Task 1 instead of allocating new components for learning these representation for the new task.

Figure 6 shows the activation of a subset of the neurons in the last hidden layer in two cases: similar tasks (sim_seq_2Tasks) and dissimilar tasks (dissim_seq_2Tasks). As illustrated in Figure 6a, there are high representational similarities between each class in Task 1 and the corresponding class that shares some semantic similarity in Task 2. On the other hand, if the tasks are dissimilar (Figure 6b), the new task has a different representation than the previous one.

**Relevant representation.** As illustrated in Figure 6, the relevant representation from previous knowledge is sparse. This representation is obtained from a sparse neural network. It would be interesting to study whether the same case holds in dense neural networks. We performed this experiment on the sim_seq_2Tasks benchmark using a **dense model** with the architecture described in Section 5.1 and the experimental setting in Appendix B. Figure 5c shows the 2D visualization of the average of the activation of the last hidden layer over samples of the class dog in Task 2 using the model trained on Task 1 $f^1$. As illustrated in the figure, even if the network is dense, the relevant representation for this new class is sparse. This is also discussed in (Sokar et al., 2021b; Javed & White, 2019).

**Backward transfer.** The main cause of catastrophic forgetting arises from updating the previously learned weights when the model is optimized for a new one. Next, we will analyze the weight change

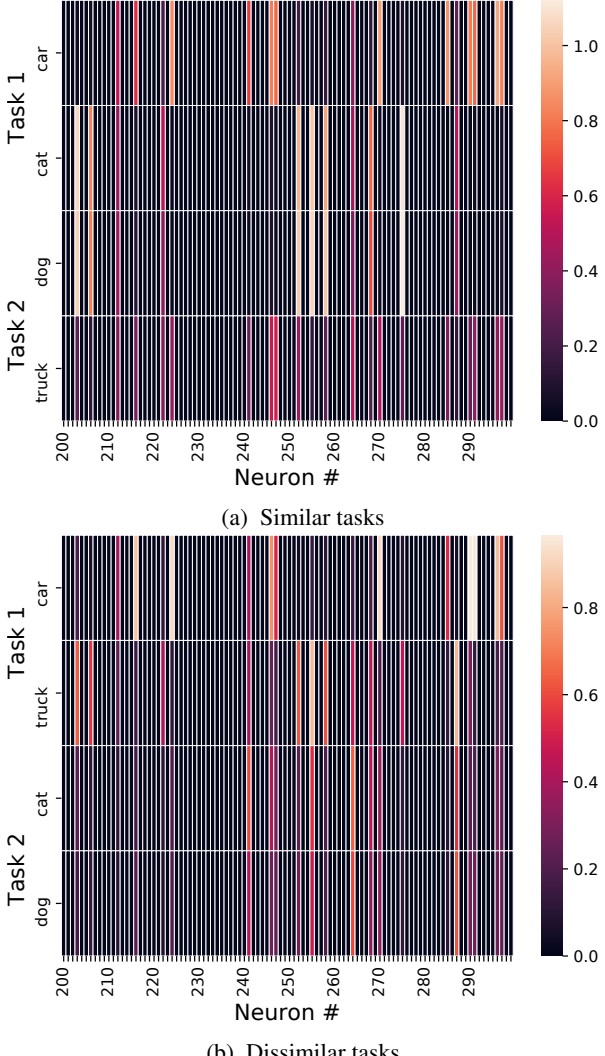

(a) Similar tasks

(b) Dissimilar tasks

Figure 6: Visualization of the representation of a subset of the neurons in the last hidden layer of each class in the sim_seq_2Tasks (a) and dissim_seq_2Tasks (b) benchmarks.

in **dense** neural networks in two cases: similar and dissimilar sequences of tasks. We performed this experiment on sim_seq_2Tasks and dissim_seq_2Tasks benchmarks. We use a dense model with the architecture described in Section 5.1 and the experimental settings provided in Appendix B. We estimated the average absolute change in the parameters in each layer between the models $f^1$ and $f^2$. This change is calculated as follows:

$$\Delta_l = \frac{1}{N_l} \sum_i (|W_{l\_i}^{t=2} - W_{l\_i}^{t=1}|), \tag{2}$$

where $W_{l\_i}^t$ is the weight $i$ in layer $l$ after learning task $t$ and $N_l$ is the number of weights in layer $l$. Figure 7 shows the average absolute change in each layer in the network. As illustrated, the change is higher when the two tasks are dissimilar. This experiment demonstrates selective update based on the relation between the classes is essential. Updating the dissimilar knowledge will cause the weights to move far away from the values learned in the previous time step. This causes forgetting of previous tasks.

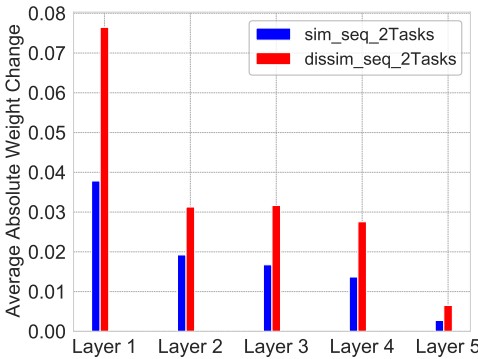

Figure 7: The average absolute difference between the weights of the model at $t = 1$ ($f^1$) and the weights at $t = 2$ ($f^2$). The change is higher when the tasks in the sequence are dissimilar.

## B EXPERIMENTAL SETUP

In this appendix, we will describe the experimental settings used in the experiments provided in Section 5.

**Datasets.** CIFAR-10 and CIFAR-100 Krizhevsky et al. (2009) are well-known benchmarks for classification tasks. They contain tiny natural images of size ($32 \times 32$). CIFAR-10 consists of 10 classes. It contains 6000 images per class (6000 training + 1000 test). While CIFAR-100 contains 100 classes, with 600 images per class (500 train + 100 test). We split these classes into tasks with different levels of semantic similarity. The network is trained using stochastic gradient descent with a batch size of 64 and a learning rate of 0.1. Each task is trained for 40 epochs. The hyperparameters are selected using a random search.

**Architecture.** The model consists of 4 convolutional layers (32-32-64-64 feature maps). The kernel size is $3 \times 3$. Max pooling layer is added after every 2 convolutional layers. Two feed-forward layers follow the convolutional layers (512-$N$ neurons), where $N$ is the total number of classes from all tasks. Similar as in (Sokar et al., 2021c), we replace the dropout layers with batch normalization (Ioffe & Szegedy, 2015).

**Hyperparameters.** Next, we will report the hyperparameters used in each experiment in Section 5. For the SpaceNet baseline, the sparsity levels $\epsilon_l$ (1-density) used in each layer is: {0.84, 0.86, 0.87, 0.87, 0.93, 0.90} for CIFAR-10 and {0.88, 0.91, 0.90, 0.97, 0.97, 0.90} for CIFAR-100. For both datasets, the number of reserved neurons for each task in each layer is: {0, 2, 2, 5, 5, 30}. Following the original paper of SpaceNet, the number of selected neurons for connections allocation $sel_l^t$ equals the number of free neurons in that layer. For KAN, on CIFAR-10, the same sparsity levels used for SpaceNet are used for the first task. For subsequent tasks, we use the above-mentioned sparsity levels starting from $l_{reuse}$. We will discuss the sparsity levels for KAN on CIFAR-100 in Appendix C.

For *Selective knowledge transfer* analysis (Section 5.2), Table 3 shows the number of candidate neurons selected for each class $sel_l^{c\text{-}cand}$ and the selected number of free neurons for each task $sel_l^t$ in each layer. We report these values for every value evaluated for $l_{reuse}$.

For *selective backward transfer* analysis (Section 5.3), the number of candidate neurons $sel_l^{c\text{-}cand}$ used in each layer from 2 to $L - 1$ is as follows: {6, 6, 15, 15, 50}.

For *longer sequence of tasks* (Section 5.4), the number of candidate neurons $sel_{l-1}^{c\text{-}cand}$ used in layers $L - 3$ and $L - 2$ are 15 and 10 respectively. While the number of selected neurons from the free list $sel_l^t$ in these two layers are 10 and 20 respectively.

## C RESOURCE EFFICIENCY AND SCALABILITY

For CIFAR-100 benchmark, sim_seq_8Tasks, we had to increase the sparsity level for each sub-network for SpaceNet to be able to fit the 8 tasks in the fixed-capacity model. The sparsity levels

Table 3: The number of the candidate neurons for each class $sel_{l-1}^{c\_cand}$ and the number of selected neurons from the free list in each layer $sel_l^t$. $x$ is the number of free neurons in layer L-1 ($|\mathbf{h}_{L-1}^{free}|$).

| | $sel_l^{c_{cand}}$ | | | | | | $sel_l^t$ | | | | | |
|---|---|---|---|---|---|---|---|---|---|---|---|---|
| $l_{reuse}/l\#$ | **1** | **2** | **3** | **4** | **5** | **6** | **1** | **2** | **3** | **4** | **5** | **6** |
| $L-1$ | 0 | 0 | 0 | 0 | 0 | 50 | 0 | 0 | 0 | 0 | 0 | 0 |
| $L-2$ | 0 | 0 | 0 | 0 | 15 | 0 | 0 | 0 | 0 | 0 | 10 | $x$ |
| $L-3$ | 0 | 0 | 0 | 15 | 10 | 0 | 0 | 0 | 0 | 10 | 20 | $x$ |
| $L-4$ | 0 | 0 | 8 | 10 | 10 | 0 | 0 | 0 | 6 | 20 | 20 | $x$ |
| $L-5$ | 0 | 8 | 6 | 10 | 10 | 0 | 0 | 6 | 12 | 20 | 20 | $x$ |
| $L-6$ | 0 | 6 | 6 | 10 | 10 | 0 | 3 | 10 | 12 | 20 | 20 | $x$ |

Table 4: The accuracy [%] of each task in sim_seq_2Tasks after learning the two tasks in sequence along with their average accuracy. The reported performance is in class-IL using different strategies for allocating the new connections in the output layer.

| Strategy for connections allocation in the last layer | $l_{reuse}$ | Task 1 | Task 2 | Average |
|---|---|---|---|---|
| Use candidate neurons only | L-1 | 69.66 | 34.77 | 52.21 |
| Use candidate neurons and free neurons | L-2 | 68.41 | 52.71 | 60.5 |
| Use free neurons only | L-2 | **64.34** | **61.11** | **62.72** |

are mentioned in Appendix B. On the other hand, since KAN utilizes the existing capacity for similar tasks, we have a free capacity that allows us to use a lower sparsity level for allocating the connections in all layers for every odd time step ($t = 1, 3, ..$). We use the same sparsity level used for CIFAR-10. For even time steps ($t = 2, 4, ..$), where we utilize the existing knowledge, we follow the same sparsity level of SpaceNet for CIFAR-100 for $l \geq l_{reuse}$.

KAN allows utilizing the resources efficiently. When the new task shares some similarities with previous existing knowledge, KAN exploits this knowledge instead of allocating unnecessary resources. The saved capacity is utilized to learn new knowledge that the agent has not encountered before. Thus, being aware of the existing knowledge helps in utilizing the available capacity efficiently and allowing for scaling to a large number of tasks in the CL sequence.

## D   CLASS AMBIGUITIES

In Section 5.2.2, we demonstrated that reusing the highest level representation ($l_{reuse} = L-1$) from the similar previous class in learning a new one causes ambiguities between the two classes in class-IL. Therefore, we suggested to: **(1)** use the free neurons only in allocating the output connections for the new class and **(2)** add new components for the new class in higher layers ($l_{reuse} < L - 1$). In Section 5.2.2, we analyzed point **(2)** by studying different values for $l_{reuse}$ while satisfying point **(1)**. In this appendix, we analyze the effect of the point **(1)**.

We performed this analysis on the sim_seq_2Tasks benchmark. We use $l_{reuse} = L - 2$. We use the candidate neurons in the output layer along with the free neurons to allocate the output connections. The number of candidate neurons $sel_l^{c\_cand}$ in layers $L - 2$ and $L - 1$ are 15 and 40 respectively. The number of selected free neurons in these two layers are 10 and 55 respectively. We compared this baseline with the KAN model presented in Section 5.2.2 with $l_{reuse} = L - 2$. The number of candidate neurons in layer $L - 2$ is 15 and **no** candidate neurons are used in layer L-1 (Table 3). We also compare against the KAN model with $l_{reuse} = L - 1$ presented in Section 5.2.1 where the connections are allocated in the last layer only using the candidate neurons in layer $L - 1$.

Table 4 shows the performance of these models. Adding new connections in earlier layers ($l_{reuse} < L - 1$) improves the performance of Task 2. However, using the free neurons only in allocating the connections in the last layer reduces the ambiguities between similar classes since each class learns its own specific representation. Thus, satisfying the **two** suggestions discussed above ($l_{reuse} < L-1$ and $\mathbf{h}_{L-1}^{alloc} = \mathbf{h}_{L-1}^{sel}$) helps in achieving the balance in performance between the tasks.

Table 5: FLOPs required to learn the sim_seq_2Tasks benchmark using different methods.

| Strategy | Method | FLOPS |
|---|---|---|
| Regularization | SI | 6.69e13 |
| Task-specific components | PackNet | 1.004e14 |
| | SpaceNet | 1.02e13 |
| | KAN (ours) | **8.14e12** |

## E    COMPUTATIONAL EFFICIENCY

In this appendix, we report the number of floating-point operations (FLOPs) required for training the regularization-based and task-specific components models. Regularization-based models are based on dense networks while task-specific components models are based on sparse networks either trained from scratch (i.e. SpaceNet (Sokar et al., 2021c), KAN (ours)) or pruned after training a dense model (PackNet) (Mallya & Lazebnik, 2018).

The FLOPs metric is the traditional metric used in the literature to estimate the speed of training algorithms and compare the computation efficiency of a sparse model against its dense counterpart (Evci et al., 2020; Yuan et al., 2021). This is because current sparse neural network models are simulated using binary masks over network weights. Truly sparse implementations with arbitrary sparsity during training is a highly researched approach with no general solution yet (Curci et al., 2021).

We follow the method described in (Evci et al., 2020) to calculate the FLOPs. The FLOPs are calculated with the total number of multiplications and additions layer by layer in the network. We compute this metric for the sim_seq_2Task benchmark.

Table 5 shows the computed FLOPs for each method. As illustrated in the table, SI requires more FLOPs than SpaceNet because it trains the dense model for each task. KAN, requires fewer FLOPs than SpaceNet as it updates the weights starting from $l_{reuse}$. PacKNet requires the highest number of FLOPs because it trains the dense model for each task and performs additional fine-tuning epochs after pruning the least important weights to recover the model performance.

## F    PERFORMANCE ACROSS SEVERAL BENCHMARKS

In this appendix, we analyze the performance of the studied task-specific components models on a sequence that contains two different datasets from two domains. We construct this sequence using two standard datasets for class-IL: MNIST LeCun (1998); Zenke et al. (2017) and Fashion MNIST (Xiao et al., 2017; Farquhar & Gal, 2019). We name the new benchmark as MNIST_FashionMNIST_6tasks. The benchmark consists of 6 tasks. Every odd task contains two consecutive classes from the MNIST dataset while every even task contains two consecutive classes from Fashion MNIST.

We performed this experiment using the same experimental setting used for the CIFAR-10 benchmark described in Appendix B except for the number of epochs. We train each task for 5 epochs. We allocate new sparse connections in each layer for Task 2 as it contains new knowledge (Fashion MNIST) different from the one learned from Task 1 (MNIST). Starting from the third task on-wards, we use $l_{reuse}$=L-3.

Figure 8 shows the average accuracy on the seen tasks at each time step. KAN succeeds in maintaining higher average performance than SpaceNet especially when the number of tasks increases. Consistent with previous findings, selective transfer of KAN is more effective than random reuse of previous components or irrelevant reuse.

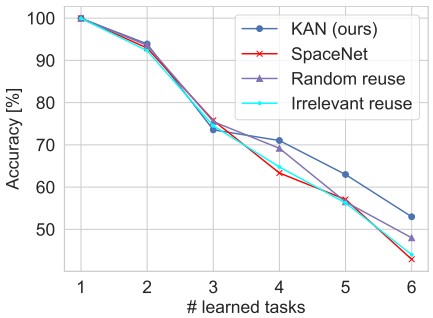

Figure 8: The performance of different methods in class-IL on MNIST_FashionMNIST_6Tasks benchmark.

# G    ADDITIONAL EXPERIMENTS AND ANALYSES

## G.1    PERFORMANCE OF REHEARSAL-BASED METHODS

In this paragraph, we compare the performance of KAN to more rehearsal-based methods. We analyze the performance of the task-specific component method PackNet (Mallya & Lazebnik, 2018). PackNet is originally designed for the task-IL as discussed in Section 2. To adapt PackNet to the class-IL, we use all the learned connections during inference without masks. The dense model is trained from scratch on the CL tasks. We prune 85% of the network weights after training each task to get approximately the same sparsity level used for the task-specific components baselines. We perform 20 training epochs for fine-tuning the network after pruning to restore its performance. We use the official code from the authors to obtain the results of PackNet on the studied benchmarks.

We also analyze the performance of other well-known regularization-based methods including EWC (Serra et al., 2018), MAS (Aljundi et al., 2018), and LWF (Li & Hoiem, 2017). We use the public code from (Masana et al., 2020) to get the results of these algorithms.

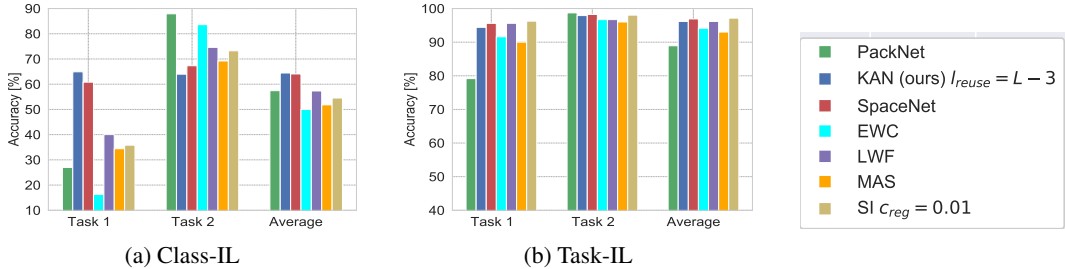

(a) Class-IL                    (b) Task-IL

Figure 9: Test accuracy of each task in sim_seq_2Tasks and their average in class-IL and task-IL.

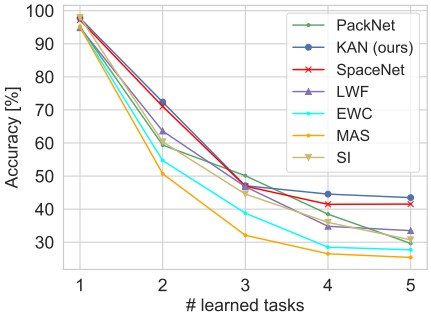

Figure 10: The performance of different methods in class-IL on sim_seq_5Tasks.

Table 6: The average accuracy (ACC) [%] and the average backward transfer (BWT) [%] on sim_seq_2Tasks and sim_seq_5Tasks benchmarks in class-IL.

| Strategy | Method | sim_seq_2Tasks | | sim_seq_5Tasks | |
|---|---|---|---|---|---|
| | | ACC | BWT | ACC | BWT |
| Regularization | SI | 54.51 | -61.6 | 30.72 | -47.66 |
| | EWC | 49.95 | - 78.60 | 27.70 | -54.90 |
| | LWF | 57.30 | - 54.90 | 33.50 | -57.4 |
| | MAS | 51.80 | - 60.50 | 25.40 | -42.3 |
| Task specific components | PackNet | 57.42 | -68.85 | 29.64 | -58.01 |
| | SpaceNet | 64.05 | -36.67 | 41.53 | -28.56 |
| | KAN (ours) | **64.43** | **-32.58** | **43.48** | **-22.46** |
| | Random reuse | 60.51 | -33.17 | 41.10 | -23.87 |
| | Irrelevant reuse | 59.25 | -29.39 | 39.54 | -26.13 |

Figure 9 and Figure 10 show the performance of the studied methods on sim_seq_2Tasks and sim_seq_5Tasks. Consistent with our previous findings, the stability-plasticity dilemma is more challenging to be addressed using regularization-based methods especially in class-IL. The methods based on Sparse representation (KAN, SpaceNet) perform better than PackNet. KAN outperforms all studied baselines on both benchmarks.

## G.2 NEGATIVE BACKWARD TRANSFER (FORGETTING)

In this paragraph, we estimate the average forgetting that occurred in the studied methods. We use the backward transfer metric (Lopez-Paz & Ranzato, 2017), BWT, which measures the influence of learning new tasks on the performance of previous tasks. Formally BWT is calculated as follows:

$$BWT = \frac{1}{T-1} \sum_{i=1}^{T-1} R_{T,i} - R_{i,i}, \qquad (3)$$

where $R_{j,i}$ is the accuracy on task $i$ after learning the $j$-th task in the sequence, and $T$ is the total number of tasks.

Table 6 shows the average accuracy over all tasks (ACC) after the model learned the whole sequence along with the average backward transfer (BWT) on the sim_seq_2Tasks and sim_seq_5Tasks benchmarks. As illustrated in the table, all regularization-based methods are more prone to forgetting than most of the task-specific components methods. PackNet is more prone to forgetting than other task-specific components methods which learn sparse representations. KAN achieves the highest average performance and least forgetting on the two benchmarks.

## H SOFTMAX CLASSIFICATION LAYER FOR CONTINUAL LEARNING

### H.1 DIFFERENT SETUPS FOR THE OUTPUT LAYER

Next, we will discuss the different setups proposed in the literature for the softmax classification layer in class-IL.

**Cosine normalization.** In (Hou et al., 2019), the authors proposed to use cosine normalization in the last layer to address the imbalance in the magnitudes of the output weights between old and new classes. Thus, the predicted probability for sample $x$ is calculated as:

$$p_i(x) = \frac{exp(\eta < \bar{W}_i, \bar{f}_{L-1}(x) >)}{\sum_j exp(\eta < \bar{W}_j, \bar{f}_{L-1}(x) >)}, \qquad (4)$$

where $\bar{f}_{L-1}(x)$ is the l2 normalized vector of the last hidden representation, $\bar{W}_i$ is the normalized weights connected to the output neuron $i$, $< \mathbf{a}, \mathbf{b} >$ is the cosine similarity between the two vectors $\mathbf{a}$ and $\mathbf{b}$, and $\eta$ is a learnable scalar to control the peakiness of softmax distribution. The setup was evaluated on a rehearsal-base method.

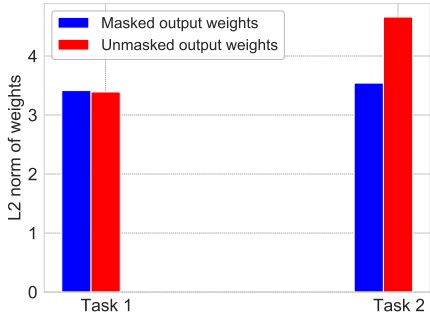

Figure 11: L2 normalization of the weights of the output layer corresponding to each task in the sim_seq_2Tasks benchmark. The magnitude of the weights of Task 2 is higher than Task 1 when the Task 1 weights are not masked during learning Task 2.

**Multi-headed output layer.** The multi-headed output setup is the common practice used in Task-IL where each task has its own output head. In (Mittal et al., 2021), the authors showed that this setup improves the performance of rehearsal-based methods in class-IL. In the multi-headed setup, the softmax activation function $\sigma(.)$ is applied on the classes of the current task only ($y \in y[t * C : t * C + C]$), where C is the number of classes in each task. The cross-entropy loss is evaluated on this selected part of the vector only.

**Pre-defined single-headed.** SpaceNet (Sokar et al., 2021c) tries to address the closed-world assumption by defining an output layer with a large number of neurons before start learning the sequence of tasks $t = 0$. To address the weights imbalance between old and new classes, only the weights of the new classes of the current task are active during their learning. During learning task $t$, a mask $M_{L-1}^t$ is used to indicate the connections connected to the output neurons for the current task where $M_{L-1}^t[t * C : t * C + C]$ equals 1 and 0 in the other indices. During learning task $t$, the weights of the output layer are multiplied by this mask which blocks the gradient flow from other output neurons:

$$W_{L-1} = W_{L-1} \otimes M_{L-1}^t, \tag{5}$$

where $\otimes$ is the element-wise multiplication operator. Figure 11 shows the effect of masking the output weights in learning a new task. We calculate the L2 normalization of the output weights of each task in the sim_seq_2Tasks benchmarks using SpaceNet. As illustrated in the figure, using all output weights in learning new tasks leads to higher weights for the new task. This causes bias in prediction towards the new task. While using a mask over the current task output weight alleviates this problem.

**Extendable layer.** This is the typical setup used in class-IL where a single-head output layer is used. At each time step, the layer is extended to include new output neurons for the new classes in the current task. In our analysis in Section 5.5, we implement the masking trick described in the previous paragraph for the extendable layer baseline. As you can see from Figure 4c, this does not solve the other limitation for the softmax being based on the closed world assumption. Figure 12 illustrates this limitation. We show the learned biases after learning two tasks from the Split CIFAR-10 benchmark in two different setups: pre-defined single-headed and extendable layer. As illustrated in the figure, the learned biases for the first task in the case of extendable layer divides the feature space between the two classes in the first task. The biases of Task 2 classes have a larger value of the bias than for Task 1. The pre-defined single-head layer overcomes this limitation.

Few previous works (Yu et al., 2020; Li et al., 2021) started to propose alternatives for the standard softmax for continual learning to overcome its limitations. More efforts are needed in addressing the limitations of the softmax to improve the performance of the CL agent in class-IL.

## H.2 RE-EVALUATION OF REGULARIZATION-BASED METHODS IN CLASS-IL

Previous studies show the huge performance drop of the regularization-based method in class-IL (van de Ven & Tolias, 2018; Hsu et al., 2018; Maltoni & Lomonaco, 2019). In these methods, the agent tends to be biased toward the last task while catastrophically forgets the previous ones.

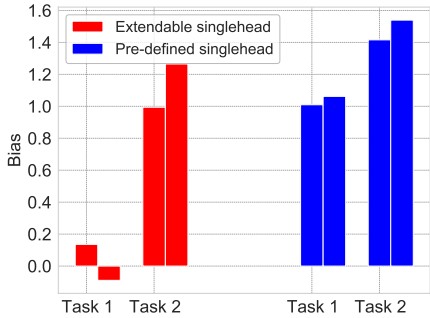

Figure 12: The learned biases in two different setups: Extendable single-headed and Pre-defined single-head. Each column represents the value of the bias for each class in the two tasks.

Table 7: Test accuracy on Split CIFAR-10 of the regularization method SI using different setups of classification output layer in the class-incremental learning.

| Setup | Accuracy |
|---|---|
| Extendable layer | $19.20 \pm 0.04$ |
| Multi-headed layer | $\mathbf{36.84} \pm 0.94$ |

We evaluated one of the regularization-based methods SI (Zenke et al., 2017) using one of the setups discussed before; multiheaded setup. We performed this experiment on the Split CIFAR-10 benchmark with 5 tasks.

Table 7 shows the average test accuracy overall tasks after learning the whole sequence. The multi-headed output layer increases the performance of the agent by 17.64%. This experiment suggests to re-consider the regularization methods in class-IL while addressing the limitations of the softmax output layer.

## I SPACENET

In this appendix, we include the details of the three main steps of the SpaceNet algorithm (Sokar et al., 2021c): (1) Connection Allocation (Algorithm 3), (2) Dynamic Sparse Training (Algorithm 4), and (3) Neuron Reservation 5. The full algorithm of SpaceNet is illustrated in Algorithm 2.

---

**Algorithm 2** SpaceNet for Continual Learning

---

1: **Require:** loss function $\mathcal{L}$ , training dataset for each task in the sequence $\mathcal{D}^t$
2: **Require:** sparsity level $\epsilon$, rewiring fraction $r$
3: **Require:** number of selected neurons $sel_l^t$, number of specific neurons $spec_l^t$
4: **for each** layer $l$ **do**
5:     $\mathbf{h}_l^{free} \leftarrow \mathbf{h}_l$                                                    ▷ Initialize free neurons with all neurons in $l$
6:     $\mathbf{h}_l^{spec} \leftarrow \emptyset$
7:     $W_l \leftarrow \emptyset$
8:     $W_L^{saved} \leftarrow \emptyset$
9: **end for each**
10: **for each** available task t **do**
11:     $\mathbf{W} \leftarrow ConnectionsAllocation(\epsilon, sel_l^t, \mathbf{h}^{free})$                           ▷ Perform Algorithm 3
12:     $W^t \leftarrow TaskTraining(\mathbf{W}, D^t, \mathcal{L}, r)$                                  ▷ Perform Algorithm 4
13:     $\mathbf{h}_l^{free} \leftarrow NeuronsReservation(spec_l^t)$                               ▷ Perform Algorithm 5
14:     $W_L^{saved} \leftarrow W_L^{saved} \cup W_L^t$                       ▷ Retain the connections of last layer for task t
15:     $W_L \leftarrow W_L \setminus W_L^t$
16: **end for each**

---

---

**Algorithm 3** Connections allocation

---

1: **Require:** number of selected neurons $sel_l^t$, sparsity level $\epsilon$
2: **for each** layer **do**
3:     $(\mathbf{h}_{l-1}^{sel}, \mathbf{h}_l^{sel}) \leftarrow$ randomly select $sel_{l-1}^t$ and $sel_l^t$ neurons from $\mathbf{h}_{l-1}^{free}$ and $\mathbf{h}_l^{free}$
4:     randomly allocate parameters $W_l^t$ with sparsity $\epsilon$ between $\mathbf{h}_{l-1}^{sel}$ and $\mathbf{h}_l^{sel}$
5:     $W_l \leftarrow W_l \cup W_l^t$
6: **end for each**

---

---

**Algorithm 4** Dynamic sparse training

---

1: **Require:** loss function $\mathcal{L}$ , training dataset $\mathcal{D}^t$, rewiring fraction $r$
2: **for each** training epoch **do**
3:     perform standard forward pass through the network parameters $\mathbf{W}$
4:     update parameters $W^t$ using stochastic gradient descent
5:     **for each** sparse parameter $W_l^t$ **do**
6:         $\widetilde{W_l^t} \leftarrow$ sort $W_l^t$ based on the connections importance $\Omega_l$
7:         $(W_l^t, k_l) \leftarrow$ drop $(\widetilde{W_l^t}, r)$            $\triangleright$ Remove the weights with smallest importance
8:         compute neuron importance $\mathbf{a}_{l-1}$ and $\mathbf{a}_l$            $\triangleright$ Neurons importance for task t
9:         $G_l \leftarrow \mathbf{a}_{l-1}\mathbf{a}_l^T$
10:        $\widetilde{G_l} \leftarrow$ sortDescending$(G_l)$
11:        $Gpos \leftarrow$ select top-$k_l$ positions in $\widetilde{G_l}$ where $W_l$ equals zero
12:        $W_l^t \leftarrow$ grow$(W_l^t, Gpos)$            $\triangleright$ Grow $k_l$ zero-initialized weights in $Gpos$
13:     **end for each**
14: **end for each**

---

---

**Algorithm 5** Neurons reservation

---

1: **Require:** number of specific neurons $spec_l^t$
2: **for each** layer $l$ **do**
3:     compute the neuron importance $\mathbf{a}_l$ for task $t$
4:     $\widetilde{\mathbf{a}}_l \leftarrow$ sortDescending$(\mathbf{a}_l)$
5:     $\mathbf{h}_l^{t_{spec}} \leftarrow$ top-$spec_l^t$ from $\widetilde{a}_l$
6:     $\mathbf{h}_l^{spec} \leftarrow \mathbf{h}_l^{spec} \cup \mathbf{h}_l^{t_{spec}}$
7:     $\mathbf{h}_l^{free} \leftarrow \mathbf{h}_l^{free} \setminus \mathbf{h}_l^{t_{spec}}$
8: **end for each**

---

## J SYNAPTIC INTELLIGENCE (SI)

SI (Zenke et al., 2017) is a regularization-based method proposed for the continual learning paradigm. It uses a fixed-capacity dense model in learning the sequence of tasks. During training each task, it estimates how important each parameter is in learning the current task. When the model faces a new task, it regularizes (penalizes) changes to parameters according to their importance to previously learned tasks. The importance of each parameter to a certain task is estimated by: (1) its contribution to the change of the loss $\omega_i^{(t)}$ (i.e. how much does a small change to the parameter change the loss function?) and (2) how far it moved $\Delta_i^{(t)}$. Following the paper notations, for the current task $t$, the per-parameter contribution in loss $\omega_i^{(t)}$ is calculated as follows:

$$\omega_i^t = - \sum_{j=1}^{N_{\text{iters}}} (\theta_i^t(j) - \theta_i^t(j-1)) \frac{\delta \mathcal{L}_j^t}{\delta \theta_i}, \tag{6}$$

where $N_{\text{iters}}$ is the total number of training iterations per task, $\theta_i^t(j)$ is the value of parameter $i$ after performing iteration $j$ on task $t$, and $\frac{\delta \mathcal{L}_j^t}{\delta \theta_i}$ is the gradient of the loss with respect to parameter $i$ at iteration $j$ during learning task $t$.

The distance travelled by a parameter $i$ during learning task $t$ is calculated as follows:

$$\Delta_i^t = \theta_i^t(N_{\text{iters}}) - \theta_i^{(t-1)}(N_{\text{iters}}). \tag{7}$$

Thus, the estimated importance of each parameter $i$ for previous $t - 1$ tasks is calculated as follows:

$$\Omega_i^{(t-1)} = \sum_{k=1}^{(t-1)} \frac{\omega_i^k}{\left(\Delta_i^k\right)^2 + \xi}, \tag{8}$$

where $\xi$ is a damping parameter to bound the expression in cases where $\Delta_i^t$ goes to zero. During learning task $t$, a regularization term $\mathcal{L}_{\text{reg}}^t$ is added to the classification loss $\mathcal{L}_{\text{cls}}^t$ to penalize the change in the weights based on their importance. $\mathcal{L}_{\text{reg}}^t$ is given by:

$$\mathcal{L}_{\text{reg}}^t = \sum_i \Omega_i^{(t-1)} \left(\theta_i - \theta_i^{(t-1)}(N_{\text{iters}})\right)^2 \tag{9}$$

The total loss for task $t$ is:

$$\mathcal{L}^t = \mathcal{L}_{\text{cls}}^t + c_{reg}\mathcal{L}_{\text{reg}}^t, \tag{10}$$

where $c_{reg}$ is a hyper-parameter to control the trade-off between the classification and regularization terms.

