# OpenReview forum: "Addressing the Stability-Plasticity Dilemma via Knowledge-Aware Continual Learning"
_ICLR.cc/2022/Conference — ICLR 2022 Submitted_

### Official Review · Reviewer_o6Js · 2021-10-29

**Correctness:** 3
**Technical Novelty And Significance:** 2
**Empirical Novelty And Significance:** 2
**Recommendation:** 3
**Confidence:** 4

**Main Review:**

Strengths:
- I really appreciated that the paper explicitly tackles the elementary questions of CL: "we investigate the stability-plasticity dilemma
to determine which model components are eligible to be reused, added, fixed, or updated to achieve this balance". Most existing methods consider a fix-size backbone and introduce new regularization or training mechanisms to avoid forgetting but do not explicitly address these problems.
Weaknesses:
- First, the method is very incremental with respect to spaceNet. In my understanding, the main difference with this previous approach lies in the lines 3 and 4 of Alg.1.
-The contribution of the paper and the difference with spaceNet should be clarified. This is discussed only in 3.2 while this should be discussed from the start in Sec.1.
-In the introduction, the second contribution is not specific to this paper. It was already in SpaceNEt
-The authors claim that they address only the class incremental learning problem. However, I don't see anything in the method that prevents the use of this method in the case of task-incremental learning. It would strengthen the experimental conclusion significantly.
-Many baselines should be added to the experiments: DEN,EWC,LWF,MAD
- The datasets used in the experiments are simple and very similar: cifar 10 or 100. I would recommend adding at least another dataset (Mnist,svhn, tinyimagenet...)
-The structure of the paper could be improved:
- Sec 3.3 should not be here. This is not part of the method. This is only the description of a baseline so I would recommend moving this part to the experimental section.
-Sec 3.2. For a reader that does not know SpaceNet, it's almost impossible to understand.
in Table1, the performances of KAN  are much worse than the other methods but it uses much fewer parameters. In my understanding, it would be possible to use different hyperparameters to have a more fair comparison allocating a more similar number of parameters? From the reported performances, the superiority of the method is not clear.
- No readme is provided with the code. So it's hard to replicate the experiments. Comments in the code are very sparse and not very informative.


detail: in figure 1 block (2), a connection is missing in the bottom-left.

**Summary Of The Paper:**

The paper address the the problem of continual learning for image classification. This approach focuses on how to reuse, add, fix or update neurons to learn tasks incrementally. The method is based on an existing method name SpaceNET.

**Summary Of The Review:**

The proposed method is based on an existing method and is very incremental. The presentation of the method could be improved. Experiments are not sufficiently convincing.

---

> ### Author Response · Authors · 2021-11-22
> **Response to Reviewer o6Js**
>
> Thank you for the thoughtful comments and for acknowledging the novelty of the problem we studied in this work. All the raised comments are addressed below and the requested experiments are performed.
>
> We would like to emphasize that the use of KAN is to perform the analyses and illustrate our findings as we stated in the last contribution point in the introduction. Thus, the goal of some experiments is to analyze some factors not to achieve the best performance.
>
> **Q1. "the method is very incremental with respect to spaceNet. In my understanding, the main difference with this previous approach lies in the lines 3 and 4 of Alg.1."**
>
> * The whole algorithm 1 is novel. Please check our answer for Q3. for the main differences between SpaceNet and KAN. Concerning the novelty, we believe that all our contributions summarized at the end of Section 1 are novel and not incremental. There are limited works that have started to study and analyze the catastrophic forgetting problem which is different from proposing a new method for solving it. More analysis is needed to understand the phenomena. To the best of our knowledge, it is the first work to deeply analyze the components of the network (reusability, fix, update,.. ) and how these affect the CL requirements (forward, backward transfer, efficiency) which is acknowledged by the reviewer. In addition, we study the gap between class-IL and task-IL and show the contribution of each of the base model and the softmax output layer in forming this gap.
>
> **Q2.“The contribution of the paper and the difference with spaceNet should be clarified”**
>
> * Thanks for the suggestion, we considered and clarified this in the revised version Section 3.2.1.
>
> **Q3. “In the introduction, the second contribution is not specific to this paper. It was already in SpaceNEt”**
>
> *The second contribution in the introduction is specific to this paper because SpaceNet doesn’t consider future tasks in connection allocation and doesn’t reuse previous components; instead, it randomly allocates new subnetwork considering only previous tasks. We have clarified this aspect in the revised version (Section 3.2.1).
>
> **Q4. “The authors claim that they address only the class incremental learning problem.”**
>
> * Our focus is to illustrate the gap in models performance between task-IL and class-IL, to study further its causes, and to analyze the stability-plasticity dilemma for CL. We didn’t claim that it is “only” for class-IL.
>
> **Q5. “Many baselines should be added to the experiments: DEN,EWC,LWF,MAD”**
>
> * **We have addressed this comment and added the requested baselines** to the revised version Appendix G. Please also refer to our answers to reviewer xDnv Q.5a) for a summary of our results.
>
> **Q6. "The datasets used in the experiments are simple and very similar: cifar 10 or 100. I would recommend adding at least another dataset (Mnist,svhn, tinyimagenet...)"**
>
> * **We follow the reviewer recommendation and add an experiment** on another benchmark which contains a mix of two datasets from different domains: MNIST and Fashion MNIST. Please check our answer to Reviewer xDnv Q.5b) for a summary. The full details can be found in Appendix F in the revised version.
>
> **Q7. "Sec 3.3 should not be here."**
>
> * Thanks for the suggestion. We moved it to the experimental section in the revised version.
>
> **Q8.a)  "Sec 3.2. For a reader that does not know SpaceNet, it's almost impossible to understand."**
>
> * We provide the complete algorithm of SpaceNet in Appendix I along with sec 3.2.
>
> **Q8.b) "in Table1, the performances of KAN are much worse than the other methods but it uses much fewer parameters…….From the reported performances, the superiority of the method is not clear."**
>
> *  Please note that the goal of the experiment in Table 1 is not to show the superiority of KAN. Instead, the goal of this experiment is to use KAN to study how the model components should be altered in two cases: dissimilar and similar sequences. We show that in the sequence of similar tasks, we can just simply reuse the existing knowledge and achieve very close performance with the other studied rehearsal-free methods. While this is not the case for dissimilar tasks, and therefore new components for the new task must be added for the case of the dissimilar tasks. The conclusion is that altering the model architecture differs and shall be a customized process based on the similarity between the tasks.
>
> **Q9. "No readme is provided with the code."**
>
> * Thanks for the comment. We will provide a readme file and more detailed comments when we will make the code public.
>
> If we have been able to clarify the main focus and contributions of the paper to the reviewer who also acknowledges that previous works “do not explicitly address these problems”, we kindly ask you to reevaluate, if possible, the score for novelty and the overall recommendation. If there are still unclear aspects, please let us know.

---

> ### Author Response · Authors · 2021-11-29
> **[Reminder] Could you please check our response?**
>
> Dear Reviewer o6Js,
>
> Thank you for your valuable comments. We would like to remind you the discussion period is ending soon. We have addressed all the raised comments and included new experiments to your concerns. Here is a summary of our response.
>
> * We provided new experiments that include comparisons with other state-of-the-art methods: PackNet, EWC, LWF, and MAS.
> * We provided new experiments on a continual sequence that has a mix of two different datasets: MNIST and Fashion MNIST.
> * We clarified the novelty and the goals of our study.
> * We clarified the difference between our study and SpaceNet.
> * We clarified the goals of the experiment presented in Table 1.
> * We improved the presentation of the paper according to the reviewer's suggestions.
>
> Would you please check them and confirm whether our response has addressed your comments?
>
> Best regards,
>
> Authors

---

### Official Review · Reviewer_UadE · 2021-11-01

**Correctness:** 2
**Technical Novelty And Significance:** 2
**Empirical Novelty And Significance:** 2
**Recommendation:** 3
**Confidence:** 5

**Main Review:**

The paper proposed a class incremental learning method that can exploit the similarity of classes for knowledge transfer. The previous work CAT did this only in the task incremental learning setting.

The paper started off quite interesting as the proposed idea is new. Limited work has been done to improve class incremental learning by transferring knowledge from previous similar classes. However, the work is incomplete as it assumes that the class similarity is given instead of being detected automatically like the CAT system. This significantly reduces the value of this work.

The experiments section has many issues. It is more like an analysis rather than showing the superiority of the proposed method. The author acknowledged this in Section 5.6.

1. Comparison with state-of-the-art class incremental learning systems is seriously lacking. Many such systems have been published recently. Practically, only SI and SpaceNet are compared, but SI is fairly old and your system is based on SpaceNet. Although existing methods do not explicitly transfer knowledge in the class-IL setting, but they do have implicit sharing and transfer because the new tasks are learned based on the model built for old tasks. Thus, a comparison with them should be included.

2. Only a fixed sequence of classes and tasks arranged by you is used. It raises the question of how you did the arrangement and what about the random sequence or simply follow the original class order in the original data. For CIFAR 100, you used only 16 classes to make 8 tasks. Why did not you use all classes in the full data to make a sequence of 50 tasks. In section 5.4, you said “every two consecutive tasks are dissimilar”. Did you arrange the order of classes to make this happen? You reported a study in figure 2 about the sensitivity of class order, but that involves only two tasks in the task-IL setting with task identity known.

3. In Section 5.2.1, you wrote “to estimate the forward transfer, we focus on the performance of Task 2 assuming the availability of the task identity at inference. Table 1 shows the accuracy of Task 2 using different methods.” Since you are working on class-IL, your assumption that the task identity is known at inference, which is the task-IL case, makes the result not reflecting the of class-IL knowledge transfer. In fact, knowledge transfer of similar classes may make the classification of the old and new classes more difficult. The question is whether this transfer in the task-IL case is beneficial to Class-IL. Also, in the results, you only reported the result of task 2. What about task 1?

4. In section 5.2.2, you reported the same experiment in the class-IL setting and showed that it does not work for class-IL and proposed two constraints. This makes the writing problematic because the constraints should have been given somewhere in the model sections as it should be part of the technique. Using only two tasks to study this is not interesting because when a large number of tasks are learned, it is very hard to know what happens.

5. Figure 4 shows that the forgetting is very serious. Many existing methods do better.

6. Section 5.2.1 and 5.2.2 are quite hard to follow as it has too many details.

7. The architecture is also very simple. Most of continual learning systems for class-IL uses ResNet.


**Summary Of The Paper:**

The paper proposed a class incremental learning method that can exploit the similarity of classes for knowledge transfer. The previous work CAT did this only in the task incremental learning setting. The paper did some analysis in the experiment section to show how the transfer can happen in some limited experiments.

**Summary Of The Review:**

This paper studies an interesting problem of knowledge transfer in the class-IL setting. Existing work only did it in the task-IL setting. However, this work is incomplete as it assumes that class similarities are known, which significantly decreases the value of this work at this stage. There are also a lot of issues in the experimental section.

---

> ### Author Response · Authors · 2021-11-22
> **Response to Reviewer UadE (1/n)**
>
> Thank you for your thoughtful comments and for recognizing the novelty of our study. Below we address all the raised comments and requested experiments.
>
> First, we would like to elaborate that the goal of this work is to provide analyses for the dynamics of the stability-plasticity dilemma. KAN is developed to ease analyzing how different altering to the model components affect the performance, forward/backward transfer, etc. Using KAN, we are able to analyze the gap between task-IL and class-IL and show that the similarity between classes plays a role in addressing different aspects for CL. The similarity on the level of classes, the case we study for class-IL, is way more challenging than the setup studied in CAT for task-IL. In addition, as we discussed in section 5.6 the similarity detection in CAT requires full training of each task from scratch which is not ideal in real-world applications, and more efforts should be devoted in this direction. We believe that our study is complete and the automatic detection of the knowledge is still an open-challenging research question that we aim to address in a separate future work.
>
> **Q1.“Comparison with state-of-the-art class incremental learning systems is seriously lacking”**
>
> * **We have included in the revised version a new comparison with other algorithms** asked by other reviewers including (PackNet, EWC, LWF, and MAS). Please see our answers to Reviewer xDnv Q.5 for a summary and Appendix G.1 in the revised version for the full details of the experiments.
>
> **Q2.a) "Only a fixed sequence of classes and tasks arranged by you is used.  how you did the arrangement”**
>
> * Our goal is to analyze how to deal with the model components (reuse, freeze, etc,) on sequences with different similarity levels. We designed the sequences in a way that forms similar, dissimilar, and a mix of tasks based on the semantic similarity.
>
> **Q2.b) “Why did not you use all classes in the full data to make a sequence of 50 tasks.”**
>
> * As the reviewer might be aware, the rehearsal-free class-IL techniques do not achieve a satisfactory level even on short sequences and there is a clear huge gap between the performance of Class-IL and Task-IL [1,2,3,4]. In this work, we aim to analyze the factors of this gap. We truly believe that closing the gap in shorter sequences will help in providing methods in the future that can do well in larger tasks as well.
>
> **Q2.c) " In section 5.4, you said “every two consecutive tasks are dissimilar”. Did you arrange the   order of classes to make this happen?"**
>
> *  In section 5.4, yes, we did arrange the order of the classes to make “every two consecutive tasks are dissimilar” happen. You can find the details in section 5.1.
>
> **Q2.d) "You reported a study in figure 2 about the sensitivity of class order, but that involves only two tasks in the task-IL setting with task identity known."**
>
> *The reason to report this in the task-IL is to show that the behavior happens even if we don’t have the additional challenges added by the class-IL (e.g. class ambiguities) for fair estimation of the importance of attention to relative knowledge.
>
> **Q3. a) "In fact, knowledge transfer of similar classes may make the classification of the old and new classes more difficult. The question is whether this transfer in the task-IL case is beneficial to Class-IL."**
>
> * This is exactly what we aim to show in this experiment as stated at the beginning of section 5.2.2 ("We showed that new tasks could be learned using previous similar knowledge in Task-IL. Does this hold in class-IL? "). Class-IL couldn’t fully utilize the existing relevant knowledge in the model due to the class ambiguities between similar classes in different tasks. To show that there is relevant knowledge in the first place (but, of course, we can’t fully use it for class-IL), we present the case with the task identity in Table 1. Then in Table 2, we show that we can’t fully utilize this knowledge providing the reasons in Section 5.2.2. Thus, the conclusion is: in task-IL, we can maximize the forward transfer of similar knowledge. However, in class-IL, maximizing the forward transfer increases the ambiguities between classes due to the absence of task-id. In the latter case, the forward transfer and negative backward transfer inevitably compete with each other. We show that the proposed constraints balance between reducing the ambiguities and increasing forward transfer.
>
> **Q3.b)  “Also, in the results, you only reported the result of task 2. What about task 1?”**
>
> * The accuracy of Task 1 can be found in Table 2.

---

> > ### Author Response · Authors · 2021-11-22
> > **Response to Reviewer UadE (2/2)**
> >
> > **Q4.a) “showed that it does not work for class-IL and proposed two constraints. This makes the writing problematic because the constraints should have been given somewhere in the model sections”**
> >
> > * Thanks for the suggestion. The idea is to describe the general version of KAN in the model section that could be applied for different scenarios. Since these constraints are needed only for the class-IL scenario, in the experiments, we first discuss the problem of class ambiguity and show that it occurs only in class-IL. Then we provide the constraints for this scenario.
> >
> > **Q4.b) " Using only two tasks to study this is not interesting"**
> >
> > * We have a different perspective on this. Studying the two tasks case is more useful for deep analysis as it represents a more controllable setting. And if the class ambiguities occur for the two tasks case, naturally, it will also occur in a larger number of tasks.
> >
> > **Q5. "Figure 4 shows that the forgetting is very serious. Many existing methods do better"**
> >
> > *  We would like to emphasize that our focus here is to study the class-incremental learning scenario using fixed-capacity models without using any form of previous data replay. We are aware of many existing methods that do better but in different settings (experience or generative reply, architecture expansion, and so on) and these are covered in the related work section. Up to our knowledge, in the realistic settings that we study, there is no method that can perform better. Following the suggestion of Reviewer NFc9, in the revised version we have added backward transfer metric (BWT) to estimate the forgetting and it can be seen that KAN has the least forgetting among all baselines (including the newly added baselines asked by the reviewers). Please see our answer to Reviewer NFc9 Q3 for a summary and Appendix G.2 for full details. If the reviewer is aware of other works that perform better, please refer them to us, and we would be happy to analyze and include them in a revised version of the paper.
> >
> > **Q6. "The architecture is also very simple”**
> >
> > * We couldn’t see why using a simpler architecture would affect our analyses. In our paper, nothing is specifically designed for simple architectures. Please, could the reviewer elaborate what are the drawbacks of using simple architecture which already has very good performance on task-IL? There is an obvious gap in the literature between task-IL and class-IL even with ResNet architectures. If the stability-plasticity dilemma is well-studied and addressed in simpler architecture, it would generalize to a more complex one. However, using more complex architectures would not directly solve the problem. For example, using ResNet for regularization methods doesn’t have a significant effect on mitigating forgetting as reported in [5] (Table 4). Addressing the challenging case of limited smaller capacity could be an added advantage to cover real-world applications with memory constraints. Still, the study is valid for both cases.
> >
> > Given the novelty of the study which is also recognized by the reviewer “The paper started off quite interesting as the proposed idea is new ..” , we kindly ask the reviewer to update his/her score for novelty. Overall, we would appreciate it if the reviewer would like to reassess the recommendation after considering our answers to all the raised comments. If something is still unclear please let us know.
> >
> > ---
> > [1] Davide Maltoni and Vincenzo Lomonaco. Continuous learning in single-incremental-task scenarios. Neural Networks, 116:56–73, 2019.
> >
> > [2] Ronald Kemker, Marc McClure, Angelina Abitino, Tyler L Hayes, and Christopher Kanan. Measuring catastrophic forgetting in neural networks. In Thirty-second AAAI conference on artificial intelligence, 2018.
> >
> > [3] Yen-Chang Hsu, Yen-Cheng Liu, Anita Ramasamy, and Zsolt Kira. Re-evaluating continual learning scenarios: A categorization and case for strong baselines. In NeurIPS Continual learning Workshop, 2018.
> >
> > [4] Sebastian Farquhar and Yarin Gal. Towards robust evaluations of continual learning. In Privacy in Machine Learning and Artificial Intelligence workshop, ICML, jun 2019
> >
> > [5] Sokar, Ghada, Decebal Constantin Mocanu, and Mykola Pechenizkiy. "Spacenet: Make free space for continual learning." Neurocomputing 439 (2021): 1-11.

---

> ### Author Response · Authors · 2021-11-29
> **[Reminder] Could you please check our response?**
>
> Dear Reviewer UadE,
>
> Thank you for your valuable comments. We would like to remind you the discussion period is ending soon. We have addressed all the raised comments and included new experiments to your concerns. Here is a summary of our response.
> * We provided new experiments that include comparisons with other state-of-the-art methods PackNet, EWC, LWF, and MAS.
> * We clarified how the benchmarks are designed and what are the goals of this design.
> * We clarified the challenges in class-IL and the big gap in performance between task-IL and class-IL.
> * We clarified that one of our goals is to show that the forward transfer is different in class-IL and task-IL despite that the knowledge in the model is the same. We presented the experiments in the order that illustrates this observation.
>
> Would you please check them and confirm whether our response has addressed your comments?
>
> Best regards,
>
> Authors

---

> > ### Comment · Reviewer_UadE · 2021-11-30
> > **Thanks for addressing my comments.**
> >
> > Thank you for addressing my comments. However, I still think the added baselines are old, published in or before 2018. Using a simple backbone architecture doesn't produce strong results. The fixed ordering of tasks and the requirement of given class similarity make this work a very preliminary investigation and little of it is conclusive.
> >
> > By the way, the name of your system KAN has been used in this paper “Continual Learning with Knowledge Transfer for Sentiment Classification.” ECML-2020.

---

> > > ### Author Response · Authors · 2021-11-30
> > > **Response to Reviewer UadE**
> > >
> > > Thanks for your response.
> > >
> > > * As asked from our first answer, please can you be more explicit and clearly tell us which Rehearsal-Free Class-IL baselines we shall add to our empirical evaluation? Please note that we have already added all baselines requested by Reviewer xDnv and Reviewer o6js.
> > >
> > > * Also, please, can you elaborate more on what is the main point of using the typical order of CIFAR-10 which is also fixed? Does this default order evaluate any aspect of an algorithm? However, in our study, we use the same classes but in a "controllable" order that allows us to study specific questions "how the model should be altered in different similarity levels?" and "can we benefit from the previous similar classes?". Thus, we have to design similar tasks, dissimilar tasks, and mix. These aspects wouldn’t be satisfied by the typical CIFAR-10 order.
> > >
> > > * The results are not strong in class-IL because of the well-known gap in the literature between task-IL and class-IL without using rehearsal strategies (such as experience or generative replay). Even if it is out of the scope of the paper, it can be seen from the empirical evaluation that our results are strong on task-IL which indicates that the chosen architecture is not a downgrading factor. Please kindly check the stated references in our previous response [1,2] where it is illustrated that even with ResNet the performance of Rehearsal-Free methods is not satisfactory for Rehearsal-Free Class-IL settings.
> > >
> > > Indeed, we made an unfortunate choice for the acronym of our proposed method. We have not been aware that the KAN acronym has been used in that paper, but on the positive side, the full name of our method (Knowledge-Aware contiNual learner) was not used. Thanks for letting us know.

---

### Official Review · Reviewer_NFc9 · 2021-11-01

**Correctness:** 4
**Technical Novelty And Significance:** 2
**Empirical Novelty And Significance:** 2
**Recommendation:** 6
**Confidence:** 5

**Main Review:**

### Strengths:
(1) The paper is well-motivated, well-written, and easy to follow.
(2) I like the overall experimental designs where different aspects of continual learning with KAN are studied with several experiments.



### Weaknesses:

Although I enjoyed the experimental design, and agree with the intuition behind KAN, the proposed method have major weaknesses:

(1) First, there are several hyper-parameters involved, such as the number of layers to be reused, the sparsity level, etc.
While the paper studies the impact of layer reuse, this is usually an unknown factor in practice for new and unseen data.

(2) Second, given all the interesting effort for developing KAN, the proposed method does not outperform existing baselines.
For instance, when the benchmark is challenging, such as Fig 4. b, KAN, SpaceNet, and even Random Reuse achieve the same performance.
I believe since the paper is an empirical work, the fact that the proposed method is not performing better than baselines hurts its contribution.

(3) Generally, while reporting metrics, it is not clear how much average forgetting for different methods are (see [1, 2] for the definition of forgetting metric). I think it is important to report these metrics as well, which indicates the negative backward transfer.


Misc:
- In Fig. 1, a fixed connection from the last neuron of the first layer is removed from steps 2 to 3.
I think there was an error in figure generation.
- It will be interesting to include the run-time of different algorithms to see how much different modifications by KAN impact the speed.

[1] Chaudhry, Arslan, et al. “Efficient Lifelong Learning with A-GEM.” International Conference on Learning Representations, 2018.
[2] Mirzadeh, Seyed Iman, et al. “Understanding the Role of Training Regimes in Continual Learning.” Advances in Neural Information Processing Systems, vol. 33, 2020, pp. 7308–7320.



-----
***UPDATE**
While I am not fully convinced regarding (1), the new version of paper addresses (2) and (3) an I now lean towards accepting this work.




**Summary Of The Paper:**

### Summary:
The paper studies replay-free continual learning with the focus on the plasticity-stability dilemma.
More specifically, the paper proposes the KAN method for class-incremental learning where the forward transfer is achieved by detecting similar knowledge and reusing the first few layers,
and negative backward transfer can be alleviated sparse connection allocations for different classes of different tasks.
The paper also studies a limitation of the softmax layer, which is an interesting contribution.

**Summary Of The Review:**

I believe the proposed method has its flaws but overall I believe it is an illustrating study and I lean towards acceptance.

---

> ### Author Response · Authors · 2021-11-22
> **Response to Reviewer NFc9**
>
> Thank you for your thoughtful comments and for recognizing the different aspects studied in the work. Below we address the raised comments.
>
> **Q1.“there are several hyper-parameters involved”**
>
> * We believe that hyper-parameters are beneficial and necessary for the continual learning setup. Since the goal of continual learning is to accumulate a large number of tasks over time and the availability of unlimited capacity couldn’t be assumed, managing the capacity used by each task through the sparsity level and number of reused layers is essential. The allocated capacity of a new task is a factor of: the total available capacity in a system, the complexity of the new task (e.g. number of new classes in the task), the similarity between the new task and previous ones, and the maximum target number of tasks if available. The proposed hyper-parameters give the facility to control and balance between the performance and memory and computation efficiency based on the above-mentioned variables of each system.  From our study, we can conclude that the number of reused layers mainly depends on the task similarities (the higher the dissimilarity is, the fewers layers should be reused) in addition to the above-mentioned factors.
>
> **Q2. "Fig 4. b, KAN, SpaceNet, and even Random Reuse achieve the same performance...the fact that the proposed method is not performing better than baselines hurts its contribution."**
>
> * We would like to elaborate that our goal here is to study the broader view of CL; addressing different aspects, as the reviewer recognized, not just the accuracy. In Fig 4.b, even if KAN has the same average accuracy of SpaceNet in the *last two time steps* in the sequence, we show that this performance could be achieved by reusing the existing knowledge learned from previous similar tasks. This reduces the capacity allocated for new similar tasks which enables us to allocate more resources to new tasks that are dissimilar to previous knowledge (See Appendix C). Hence, we utilize the available limited capacity efficiently. Moreover, KAN is less prone to forgetting than SpaceNet on this benchmark. Average BWT of KAN: -26.78 while BWT of SpaceNet is: - 28.48.  We don’t believe that KAN hurts the contributions, instead, it shows that we can address other requirements and desiderata for continual learning (e.g. forward/backward transfer, efficiency) while maintaining the same performance.
>
> **Q3. " I think it is important to report these metrics as well, which indicates the negative backward transfer."**
>
> * Thank you very much for this suggestion. We agree with the reviewer that adding the forgetting metric gives a clear measure, but we overlooked it in the initial submission. **We added Backward Transfer (BWT) metric** in the revised version Appendix G.2 and a summary is provided below.
> |  |Sim_seq_2Tasks| | | Sim_seq_5Tasks||
> |---|---|---|---|---|---|
> Method | ACC [%] | BWT [%] |   |ACC [%]| BWT [%]|
> SI creg=0.01| 54.51| -61.60| |30.72 | -47.66
> EWC | 49.95 | -78.60 || 27.70 | -54.90
> LWF | 57.30 | -54.90 | |33.50 | -57.4
> MAS | 51.80|-60.50||25.40|-42.30
> PackNet | 57.42|-68.85|| 29.64|-58.01
> SpaceNet| 64.05 | -36.67| |41.53|-28.56
> KAN L-3 (ours) | **64.43** | **-32.58** | |**43.48** | **-22.46**
> Irrelevant reuse | 59.25 | -29.39 || 39.54 | -26.13
> Random reuse | 60.51 | -33.17 || 41.10 | -23.87
>
>
> **Misc:**
> 1. Thanks for pointing this out. We resolved it in the revised version.
> **2. "It will be interesting to include the run-time of different algorithms"**
> * **We added the number of FLOPs** as a measure of the running time in Appendix E in the revised version and a summary can be found below for sim_seq_2Tasks. Please note that FLOPs are the typically used measure in the literature for estimating the speed of training algorithms using sparse neural network models [1,2]. This is because current sparse neural network models are simulated using binary masks over network weights. Truly sparse implementations with arbitrary sparsity during training is a highly researched approach with no general solution yet [3] and implementing it would be a research topic in itself and out of the scope of this paper.
>
> | Method | FLOPs|
> |---|---|
> |SI | 6.69e13|
> |SpaceNet | 1.02e13|
> |KAN L-3 (ours) | **8.14e12** |
> |Irrelevant reuse | 8.14e12 |
> |Random reuse | 8.14e12 |
> |PacKNet | 1.004e14|
>
> ---
> [1] Evci, Utku, et al. "Rigging the lottery: Making all tickets winners." International Conference on Machine Learning. PMLR, 2020.
>
> [2] Yuan, Geng, et al. "MEST: Accurate and Fast Memory-Economic Sparse Training Framework on the Edge." Thirty-Fifth Conference on Neural Information Processing Systems. 2021.
>
> [3] Curci, Selima, Decebal Constantin Mocanu, and Mykola Pechenizkiyi. "Truly Sparse Neural Networks at Scale." arXiv preprint arXiv:2102.01732 (2021).

---

> ### Author Response · Authors · 2021-11-29
> **[Reminder] Could you please check our response?**
>
> Dear Reviewer NFc9,
>
> Thank you for your valuable comments. We would like to remind you the discussion period is ending soon. We have addressed all the raised comments and included the requested metrics in our experiments. Here is a summary of our response.
> * We included the forgetting metric in our experiments and analyses.
> * We included the FLOPs to analyze and estimate the running time required by each strategy.
> * We clarified the importance of the hyperparameters for the continual learning paradigm.
> * We clarified the multiple aspects we study in KAN (e.g. Forward transfer, Backward transfer, reusability of existing components, memory, computation, etc).
>
> Would you please check them and confirm whether our response has addressed your comments?
>
> Best regards,
>
> Authors

---

> > ### Comment · Reviewer_NFc9 · 2021-11-30
> > **Response to the authors**
> >
> > I want to thank the authors for addressing my comments, and I believe the paper is in better shape now. For that, I will raise my score. However, while I agree that the hyper-parameters give flexibility, I still stand by my argument that this hurts KAN in practical settings as it is extremely difficult to estimate the complexity of new data, and in online settings, this is not feasible. Overall, with the recent improvements, the paper is closer to acceptance, in my opinion.

---

### Official Review · Reviewer_xDnv · 2021-11-03

**Correctness:** 3
**Technical Novelty And Significance:** 3
**Empirical Novelty And Significance:** 3
**Recommendation:** 6
**Confidence:** 3

**Main Review:**

Pros:
+ This work is well motivated. Exploring similarity between tasks could be indeed a promising way to address the stability-plasticity dilemma to simultaneously promote forward transfer and avoid catastrophic forgetting.
+ Extensive ablation studies are conducted to show the proposed method work to some extent.

Cons:
- The organization of method section could be improved.
For example, the first and second steps could be divided into two separate paragraphs, and selective gradient back-propagation could become another paragraph.
In addition, it is better to summarize the categorization of neurons in a layer.
- There is no explanation on why activation of a neuron could be a good indicator for relevance between classes. What else do authors also attempt?
- It is suggested to calculate the statistics about how much portion of neurons are reserved, reusable and free for switching between similar tasks, and how much for the switch between dissimilar ones.
- The proposed method also shares some similarity with PackNet although the PackNet does not explore similarity between tasks but tries to compress the used neurons for each new task. Authors should compare the proposed method with it to see which method is more effective in address the stability-plasticity dilemma.
[a] A. Mallya and S. Lazebnik, PackNet: Adding Multiple Tasks to a Single Network by Iterative Pruning, CVPR 2018.
- More rehearsal-free CL methods should be compared with. In addition, comparison on larger datasets such as tinyImageNet and also the one across several datasets such as Oxford Flowers, Caltech-UCSD Birds and MIT Scenes should be conducted.

**Summary Of The Paper:**

This work aims at addressing the stability-plasiticity dilemma in the class incremental learning scenario by exploring which model components shold be reused, added, fixed or updated. Authors proposed to make use of existing knowledge in previous tasks, including identifying and adding similar knowledge that could be reusable for forward transfer and preventing dissimilar knowledge from being transferred to avoid forgetting, hence achieving better balance between catastrophic forgetting and forward transfer. Extensive ablation studies are conducted to demonstrate the effectiveness of the proposed method.

**Summary Of The Review:**

Good motivation and extensive ablation study. But the method should be described in more details with more insightful explanation in a better organizing way. In addition, it should be compared with more rehearsal-free methods on larger datasets.

---

> ### Author Response · Authors · 2021-11-22
> **Response to Reviewer xDnv (1/n)**
>
> Thank you for your valuable feedback. We are glad to see that the motivation of the study is recognizable and promising for addressing *multiple* continual learning desiderata simultaneously. Please find our answers to the raised comments and requested experiments below and in the revised version.
>
> **Q1. “The organization of the method section could be improved.”**
> * Thanks for the suggestion. We improved this part in the revised version.
>
> **Q2. “There is no explanation on why activation of a neuron could be a good indicator for relevance between classes. What else do authors also attempt?”**
> * Neuron activity is a well-known metric in the pruning literature for estimating the importance of a neuron/connection in a model [1,2,3]. We used this metric here to select the important parts from the previously learned knowledge. Figure 6 gives an intuition of the activation in the case of similar and dissimilar tasks. As illustrated in this figure, in the dissimilar case different neurons are active. We assessed this metric by comparing it to the two proposed baselines: random and irrelevant reuse. In the future, we aim to include an automatic attention mechanism for selecting relevant knowledge. The motivation for using the simple yet effective approach here, neuron activity, is to avoid adding extra factors (e.g. forgetting in attention modules) that would affect and bias our analysis of the main model components.
>
> **Q3. "It is suggested to calculate the statistics about how much portion of neurons are reserved, reusable and free for switching between similar tasks, and how much for the switch between dissimilar ones.”**
> * These details are provided in Appendix B. A summary of the statistics for the sim_seq_2task and dissim_seq_2task is provided below. We calculate the total number of reuse, reserved, free neurons in all layers with respect to the total number of neurons in the network. Please note that the lower-level layers in the network have fewer neurons than the higher-level ones. This is reflected in the small percentage of reuse. In addition, we reserve a small number of neurons as stated in Appendix B since the model capacity is small.
>
> |Benchmark|  Resue | Reserve  | Free |
> |-----|---|---|---|
> sim_seq_2task|18.52%|31.10%|68.89%
> dissim_seq_2task|0%|32.38%|67.61%
>
>
> **Q4. "The proposed method also shares some similarity with PackNet. Authors should compare the proposed method with it to see which method is more effective in address the stability-plasticity dilemma."**
> * PackNet is designed for the task-IL setting as it requires the task-id to select the corresponding mask of a task as discussed in Section 2 while our main focus here is on class-IL. Nevertheless, **we followed the suggestion of the reviewer and add an additional experiment to include PackNet in our analyses**, adapted to class-IL by using all the learned connections at inference without masks as in SpaceNet and KAN. The details can be found in Appendix G.1 in the revised version. A summary of the results can be found in our answer to  Q5 a).

---

> > ### Author Response · Authors · 2021-11-22
> > **Response to Reviewer xDnv (2/2)**
> >
> > **Q5. a) “More rehearsal-free CL methods should be compared with.”**
> >
> > * **We have compared our method with the additional 3 rehearsal-free methods** suggested by Reviewer o6Js along with PackNet. Please find the summary below and more details in Appendix G.
> >
> > sim_seq_2Tasks.  *ACC stands for accuracy.
> >
> > |Method|   | Class-IL | |    | Task-IL |  |
> > |-----|---|---|---|---|---|---|
> > | |Task 1 | Task 2 | Average ACC [%] | Task 1| Task 2 |Average ACC [%]||
> > PackNet [4] | 26.95 | 87.90 | 57.42 | 79.15 | 98.7 | 88.92
> > EWC [5] | 16.3 | 83.60 | 49.95 | 91.6 | 96.7 | 94.15
> > LWF [6] | 40.0 | 74.60 | 57.3 | 95.6 | 96.7 | 96.15
> > MAS [7] | 34.4 | 69.20 | 51.8 | 90.0 | 96.0 | 93.0
> > SpaceNet | 60.80 | 67.30 | 64.05 | 95.59 | 98.23| **96.90**
> > KAN reuse=L-3 (ours) | 64.90 | 63.97 | **64.43** | 94.41 | 97.90 | 96.15
> >
> > sim_seq_5Tasks (ACC) [%]
> >
> > | # encountered tasks| 1  | 2  | 3  | 4  | 5  |
> > |-----|---|---|---|---|---|
> > |PackNet [4] | 95.1 | 59.37 | 50.11 | 38.48 | 29.64|
> > |EWC [5] | 95.0 | 54.7 | 38.8 | 28.5 | 27.7|
> > |LWF [6] | 94.85 | 63.62 | 46.78 | 34.82 | 33.50|
> > |MAS [7] | 95.0 | 50.70 | 32.10 | 26.50 | 25.40|
> > |KAN L-3 (ours) | 97.77 | 72.38 | 47.07 | 44.55 | **43.48**|
> >
> > **Q5. b) “a comparison across several datasets.”**
> >
> > * **We added a new experiment across two commonly used benchmarks for batch class-IL**: MNIST and Fashion MNIST which is also suggested by Reviewer o6Js. We construct a sequence of 6 tasks. Every odd task contains two classes from the MNIST dataset. While every even task contains two classes from the Fashion MNIST dataset. We named this benchmark as MNIST_FashionMNIST_6Tasks. KAN starts to reuse the existing knowledge after the model learns one task from MNIST and another task from FashionMNIST (i.e. starting from t=3). We use lreuse=L-3. Below is the average accuracy at each time step. The full details of the experiment can be found in Appendix F in the revised version.
> > Please note that we let the other suggested benchmarks for future work due to the lack of time and computational resources during the short rebuttal period.
> >
> > MNIST_FashionMNIST_6Tasks (ACC)
> >
> > | # encountered tasks| 1  | 2  | 3  | 4  | 5  | 6  |
> > |-----|---|---|---|---|---|---|
> > |  SpaceNet | 99.95 | 92.92 | 75.80 | 63.34 | 57.08 | 42.91   |
> > |    KAN L-3 (ours) | 99.95  | 93.89 | 73.55 | 71.03 | 62.98 | **52.98**  |
> > |    Random Reuse | 99.95 | 93.57 |  75.5 | 69.19 | 56.51 | 48.0  |
> > |    Irrelevant reuse | 99.95 | 92.30 | 74.507 | 64.769 |56.26 | 44.01|
> >
> > ---
> > [1] Hu, Hengyuan, et al. "Network trimming: A data-driven neuron pruning approach towards efficient deep architectures." arXiv preprint arXiv:1607.03250 (2016).
> >
> > [2] Luo, Jian-Hao, Jianxin Wu, and Weiyao Lin. "Thinet: A filter level pruning method for deep neural network compression." Proceedings of the IEEE international conference on computer vision. 2017.
> >
> > [3] Dekhovich, Aleksandr, et al. "Neural network relief: a pruning algorithm based on neural activity." arXiv preprint arXiv:2109.10795 (2021).
> >
> > [4] A. Mallya and S. Lazebnik, PackNet: Adding Multiple Tasks to a Single Network by Iterative Pruning, CVPR 2018
> >
> > [5] Kirkpatrick, James, et al. "Overcoming catastrophic forgetting in neural networks." Proceedings of the national academy of sciences 114.13 (2017): 3521-3526.
> >
> > [6] Li, Zhizhong, and Derek Hoiem. "Learning without forgetting." IEEE transactions on pattern analysis and machine intelligence 40.12 (2017): 2935-2947.
> >
> > [7] Aljundi, Rahaf, et al. "Memory aware synapses: Learning what (not) to forget." Proceedings of the European Conference on Computer Vision (ECCV). 2018.

---

> ### Author Response · Authors · 2021-11-29
> **[Reminder] Could you please check our response?**
>
> Dear Reviewer xDnv,
>
> Thank you for your valuable comments. We would like to remind you the discussion period is ending soon. We have addressed all the raised comments and provided new experiments to your concerns. Here is a summary of our response.
> * We provided a comparison with PackNet.
> * We provided new experiments that include comparisons with more rehearsal methods namely EWC, LWF, MAS.
> * We provided new experiments on a continual sequence that has a mix of two different datasets: MNIST and Fashion MNIST.
> * We improved the organization of the method section.
> * We clarified the reason for using the activation to identify the relation between classes and provided the requested statistics on reserved and free neurons.
>
> Would you please check them and confirm whether our response has addressed your comments?
>
> Best regards,
>
> Authors

---

### Decision · Program_Chairs · 2022-01-20

**Decision:**

Reject

**Comment:**

This paper investigates the stability-plasticity dilemma in the class incremental learning context. It investigates which model components are eligible to be “reused, added, fixed, or updated” to achieve a good balance. Initially the paper had one supporter (xDnv) who liked the motivation and extensiveness of the ablation. NFc9 and UadE also echoed some similar points about motivation and liked that the work was easy to follow. Reviewers expressed concerns such as incrementality w.r.t. spaceNet, too many hyper-parameters, unclear performance benefit, lack of comparison to SOTA, fixed sequence of classes specified by the authors, not clear how much forgetting is happening in each method (echoed by multiple reviewers), and limited datasets used for evaluations. The authors responded to the critical reviews and provided a revised version of the paper with additional comparisons to rehearsal-free methods and with more datasets (MNIST/FashionMNIST).

Following the author response, NFc9 stated that they thought the paper was in better shape with the revisions and upgraded their score claiming it was “closer to acceptance”. Yet, they still had concerns with the practical implications of having too many hyper-parameters. UadE engaged further with the authors but claimed that they avoided the reviewer’s direct concerns. UadE maintained their concerns with the manual ordering of classes and older baselines. I agree that there are several rehearsal or pseudo-rehearsal methods to which they could have compared. Reviewer o6Js did not engage further. Overall this is a borderline paper. I appreciate the authors engaging in the discussion period, though my assessment is that the key issues still remain. This paper could use further development so my recommendation is reject.